# CHAIN-OF-CONTEXT LEARNING: DYNAMIC CONSTRAINT UNDERSTANDING FOR MULTI-TASK VRPS

**Shuangchun Gui[1], Suyu Liu[2], Xuehe Wang[3], Zhiguang Cao[1]***

[1]School of Computing and Information Systems, Singapore Management University, Singapore
[2]Guangdong Laboratory of AI and Digital Economy, Shenzhen, Guangdong, China
[3]School of Artificial Intelligence, Sun Yat-sen University, Zhuhai, Guangdong, China
`gshuangchun@outlook.com, liusuyu97@gmail.com,`
`wangxuehe@mail.sysu.edu.cn, zgcao@smu.edu.sg`

## ABSTRACT

Multi-task Vehicle Routing Problems (VRPs) aim to minimize routing costs while satisfying diverse constraints. Existing solvers typically adopt a unified reinforcement learning (RL) framework to learn generalizable patterns across tasks. However, they often overlook the constraint and node dynamics during the decision process, making the model fail to accurately react to the current context. To address this limitation, we propose *Chain-of-Context Learning* (CCL), a novel framework that progressively captures the evolving context to guide fine-grained node adaptation. Specifically, CCL constructs step-wise contextual information via a Relevance-Guided Context Reformulation (RGCR) module, which adaptively prioritizes salient constraints. This context then guides node updates through a Trajectory-Shared Node Re-embedding (TSNR) module, which aggregates shared node features from all trajectories' contexts and uses them to update inputs for the next step. By modeling evolving preferences of the RL agent, CCL captures step-by-step dependencies in sequential decision-making. We evaluate CCL on 48 diverse VRP variants, including 16 in-distribution and 32 out-of-distribution (with unseen constraints) tasks. Experimental results show that CCL performs favorably against the state-of-the-art baselines, achieving the best performance on all in-distribution tasks and the majority of out-of-distribution tasks.

## 1 INTRODUCTION

The vehicle routing problem (VRP) seeks to determine optimal routes for a fleet of vehicles to serve a set of customers while satisfying operational constraints such as vehicle capacity. Efficiently solving VRPs can significantly reduce transportation costs and improve service quality, making it a critical task in logistics and supply chain management (Toth and Vigo, 2014; Konstantakopoulos et al., 2022; Garaix et al., 2010; Dondo et al., 2011). Traditional approaches (Perron and Furnon; Lin and Kernighan, 1973; Vidal et al., 2020) often rely on heuristic-based solvers, such as LKH (Lin and Kernighan, 1973) and HGS (Vidal et al., 2020). While effective in certain settings, these methods are computationally intensive and typically require extensive hand-crafted rules to adapt to different problem variants. Recently, neural networks have emerged as a promising alternative due to their flexibility and ability to learn generalizable policies (Joshi et al., 2019; Kool et al., 2019; Kwon et al., 2020; Wu et al., 2021; Ma et al., 2023; Sun and Yang, 2023; Bengio et al., 2021; Bogyrbayeva et al., 2024; Hottung and Tierney, 2020; Hottung et al., 2022; Xin et al., 2021; Chalumeau et al., 2023; Ma et al., 2023; Chen et al., 2023a). These neural solvers are trained offline using historical or synthetically generated instances, enabling fast inference at test time for a given VRP variant. However, real-world VRPs often involve more complex and diverse constraints beyond vehicle capacity, leading to multi-task VRPs, where each task involves a different combination of constraints. This makes the neural VRP solvers for a specific single task less effective due to the massive yet necessary re-training or fine-tuning.

---

*Corresponding author.

In multi-task VRPs, the commonly studied constraints include *backhaul demands* (B) (Zong et al., 2022; Kong et al., 2024), *open routes* (O) (Tyasnurita et al., 2024; Bezerra et al., 2023), *route duration limits* (L) (Oliveira et al., 2025), *customer time windows* (TW) (Zhang et al., 2022; Lin et al., 2021), *mixed backhaul* (MB) (Wang et al., 2024), and *multi-depot settings* (MD) (Karakatič and Podgorelec, 2015). To tackle the multi-task scenario, a number of neural models (Liu et al., 2024; Zhou et al., 2024a; Berto et al., 2024; Li et al., 2025) have been developed using a unified reinforcement learning (RL) framework, which encodes both constraint information and node attributes into static embeddings. The decoding stage follows a Markov Decision Process (MDP). For a given VRP task, the model combines a global context, such as current time or remaining vehicle capacity, with these static node embeddings to select the next node. Since node priorities change across decoding, static node embeddings, which remain fixed across decoding steps, cannot reflect this dynamic property. While the context is updated, such a misaligned context-node pair may lead to inaccurate state estimation, thereby misjudging the next decision.

To overcome this limitation, we argue that ***constraint requirements should be explicitly integrated into the step-wise context and used to adaptively refine node-level representations***. In single-task VRPs, dynamic decoding mechanisms, such as the removal of visited nodes (Xin et al., 2020), have been used to reflect evolving routing decisions. While conceptually related, extending such a mechanism to multi-task settings introduces three unique challenges: (1) The importance of each constraint may vary across decoding steps, *e.g.*, the open route constraint becomes more critical as a vehicle's sub-route nears completion. Applying uniform attention across all constraints at each step, such as the one in (Li et al., 2025), limits the model's ability to focus on the most important ones. Moreover, performing RL-based node refinement into VRPs poses issues with efficiency and sequential dependencies. On the one hand, (2) multi-trajectories involve different contexts at each step, and re-embedding the nodes for each context (*e.g.*, (Xin et al., 2020)) causes a heavy computational burden. On the other hand, (3) multi-task VRP solvers (Li et al., 2025; Berto et al., 2024; Zhou et al., 2024a) typically refine the node representations at step-$i$ using only the initial (step-0) embeddings and the current context. A misaligned state may fail to capture the status of the current decoding step, thereby limiting the model's ability to accurately represent the Markov property, which is essential for coherent sequential decision-making.

To address these challenges, we propose *Chain-of-Context Learning* (CCL), a novel framework for constraint-aware, step-wise reasoning in multi-task VRPs. Specifically, to tackle *Challenge (1)*, CCL constructs step-wise contextual information using a Relevance-Guided Context Reformulation (RGCR) module. RGCR combines constraint-specific attributes (*e.g.*, remaining capacity for B and current time for TW), and adaptively emphasizes each constraint according to its similarity to the current node embedding. To address *Challenge (2)*, we design a Trajectory-Shared Node Re-embedding (TSNR) module, which enables efficient refinement of node features. TSNR employs shared node embeddings as queries and uses multi-trajectory contexts as keys and values in a multi-head attention mechanism, avoiding redundant re-embedding for each trajectory. To resolve *Challenge (3)*, TSNR updates node embeddings in the environment and feeds them as queries to the next decoding step. This design allows CCL to capture sequential dependencies and model the evolution of node importance over time.

We evaluate CCL on the combinations of six core constraints (B, O, L, TW, MB, MD), resulting in 16 in-distribution and 32 out-of-distribution multi-task VRP variants. Our contributions are summarized as follows: (1) *Conceptually*, we correct a misalignment in prior VRP formulation, by learning step-wise context and node status for a more accurate state. (2) *Methodologically*, we propose RGCR to integrate constraint requirements into the step context, along with TSNR to facilitate effective refinement and capture sequential dependencies. (3) *Experimentally*, our method achieves superior results on all seen (in-distribution) tasks and the majority of unseen (out-of-distribution) tasks.

## 2 PRELIMINARIES

**Problem Definition.** The classical vehicle routing problem (VRP) aims to determine a set of sub-routes that minimize total travel cost while satisfying customer demands. In each sub-route, a vehicle departs from the depot, delivers goods to a subset of customers, and returns to the depot, subject to the following standard constraints: (1) each sub-route starts and ends at the depot; (2) each customer is visited exactly once; and (3) the total demand on each sub-route does not exceed the vehicle's

capacity. Formally, the problem is defined on a graph where the set of nodes $\mathbf{V} = \{v_0, v_1, \ldots, v_N\}$ represents the depot ($v_0$) and $N$ customer locations. Each customer node $v_i$ is associated with a demand value $\delta_i \in [0, Q]$, where $Q$ denotes the vehicle's capacity.

Following (Berto et al., 2025), we extend this classical setting by considering six additional constraints commonly studied in multi-task VRPs: (1) **Open Routes (O)**: In problems like OVRP, this constraint is denoted by a binary flag $o \in \{0, 1\}$, which defines whether a route must return to the depot. When $o = 1$, vehicles are not required to return to the depot after completing their route. (2) **Duration Limits (L)**: In problems like VRPL, this constraint enforces a maximum route length $l$ to promote workload balancing across sub-routes. (3) **Backhaul Demands (B)**: In problems like VRPB, customer nodes are classified into linehaul and backhaul types. The vehicle must first complete all linehaul deliveries (goods from depot to customers) before collecting backhaul items (goods from customers to depot). Each customer has two types of demand: $\delta_i^l$ for linehaul and $\delta_i^b$ for backhaul, with $\delta_i^l, \delta_i^b \in [0, Q]$. (4) **Mixed Backhaul (MB)**: In problems like VRPMB, this constraint relaxes the linehaul-before-backhaul requirement, allowing both types of customers to appear in any order along the route, while still respecting the capacity constraint. (5) **Time Windows (TW)**: In problems like VRPTW, each customer $v_i$ is associated with an early time $t_i^e$, a late time $t_i^l$, and a service duration $t_i^s$. Vehicles must arrive before $t_i^l$ and wait if they arrive earlier than $t_i^e$, ensuring service occurs within the specified window. (6) **Multi-Depot (MD)**: In problems like MDVRP, this constraint allows multiple depot nodes instead of a single depot. Vehicles may begin their routes from any depot in the set, introducing additional complexity in depot assignment.

**Markov Decision Process for Multi-Task VRPs.** The multi-task VRP solver acts as a single agent, using the encoder-decoder architecture as its policy network. The policy generates a node sequence autoregressively, using a Markov Decision Process (MDP) environment

$$\mathcal{M} = (\mathcal{S}, \mathcal{A}, \mathcal{P}, \mathcal{R}). \tag{1}$$

(1) **State ($\mathcal{S}$)** consists of node embeddings and context embeddings. During decoding, following (Liu et al., 2024; Zhou et al., 2024a; Berto et al., 2024; 2025; Li et al., 2025), the model explores from diverse starting points, forming multiple trajectories in parallel. Each trajectory maintains its own context (*e.g.*, current time and used capacity), while all trajectories share the same set of node embeddings.

(2) **Action ($\mathcal{A}$)** corresponds to selecting the next node to visit. The policy network takes the current state as input and generates a trajectory-specific probability distribution over feasible nodes, allowing each trajectory to independently select its next action based on the predicted probabilities.

(3) **Transition ($\mathcal{P}$)** updates the environment after a node is selected. This modifies the environmental routing information, such as the vehicle's current position and remaining capacity. The updated environment then defines the next context embedding and continues the decision process.

(4) **Reward ($\mathcal{R}$)** is defined as the negative total route length. After all nodes are visited, each trajectory computes its own negative route length as the reward. These rewards, together with the action log-probabilities produced by the policy network, are aggregated to form a single training objective. The policy network parameters $\theta$ are then updated using the REINFORCE gradient (Williams, 1992):

$$\nabla_\theta J(\theta) = \frac{1}{N} \sum_{i=1}^{N} (R_i - b) \nabla_\theta \log \pi_\theta(a_i \mid s_i), \tag{2}$$

where $i$ is the index of the trajectory and $\pi_\theta(a_i \mid s_i)$ denotes the probability assigned to action $a_i$ conditioned on state $s_i$. $b$ is a shared baseline used to reduce gradient variance, computed as the average reward over all trajectories.

## 3 METHODOLOGY

Existing works only update the context embeddings while keeping node embeddings fixed. As described in Section 2, the current state should include both candidate node embeddings and context embeddings. In our method, we treat context and node status as a pair, ensuring that both reflect the status of the current decoding step. During environment updates, we update both simultaneously to maintain alignment between context and node information.

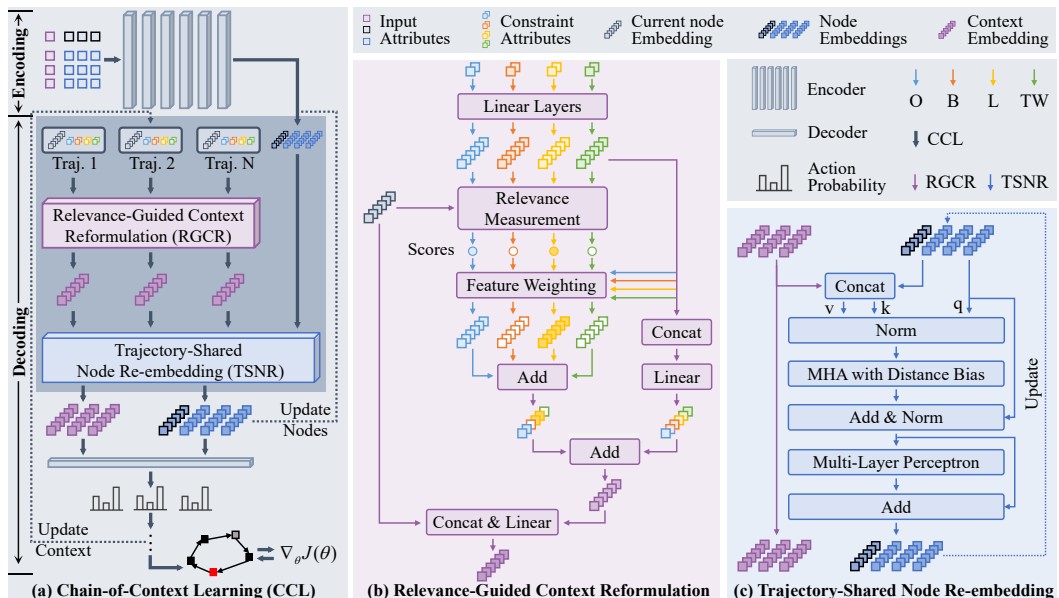

Figure 1: (a) CCL enables fine-grained constraint understanding by integrating RGCR and TSNR into the decoding stage. (b) and (c) illustrate the internal architectures of RGCR and TSNR, respectively.

## 3.1 OVERVIEW OF CHAIN-OF-CONTEXT LEARNING (CCL)

Fig. 1 (a) illustrates the training workflow of our proposed *Chain-of-Context Learning* (CCL). It adopts the classic encoder-decoder paradigm, with Relevance-Guided Context Reformulation (RGCR) and Trajectory-Shared Node Re-embedding (TSNR) integrated into the decoding stage. During encoding, each VRP instance-comprising constraints, depot, and node features-is embedded using a transformer encoder. Instances from 16 tasks, derived from the four constraints (B, L, O, and TW), are combined into a single batch for multi-task learning. In the RL-based decoding stage, CCL employs a lightweight architecture to make decisions, with multiple trajectories explored in parallel from diverse starting points. At each decision step, RGCR aggregates the constraint-specific attributes and current node embedding to generate a context embedding. After collecting the context embeddings from all trajectories, TSNR refines the historical node embeddings by jointly processing them with the multi-trajectory contexts. These refined node embeddings are passed to the next step, progressively influencing context construction and forming a Chain-of-Context across decoding steps. The constructed context and refined node features are used together to make the decision, with all components jointly optimized using an RL objective. The inference procedure is similar to the training setup, except it is extended to evaluate generalization on two additional constraints, *i.e.*, MB and MD, which are held out during training for zero-shot evaluation.

## 3.2 ENCODER

In the encoding stage, as shown in Fig. 1 (a), the inputs includes the contraint flag $\tilde{\mathbf{h}}$ and the node attributes $\mathbf{h} = \{\mathbf{h}_0, \mathbf{h}_1, \ldots, \mathbf{h}_N\}$. These attributes are embeded through a transformer-based encoder $\mathcal{E}(\cdot)$, resulting in node embeddings $\mathbf{H} \in \mathbb{R}^{(N+1) \times D}$:

$$\mathbf{H} = \mathcal{E}(\tilde{\mathbf{h}}, \mathbf{h}). \tag{3}$$

Following (Li et al., 2025), the constraint label $\tilde{\mathbf{h}} \in \mathbb{R}^4$ is a one-hot vector to indicate the presence of 4 constraints (*i.e.*, B, O, L, TW). The depot attribute $\mathbf{h}_0 = \{c_0^x, c_0^y, o, l\} \in \mathbb{R}^4$ includes the depot coordinates, and labels of O and L. $\{\mathbf{h}_1, \mathbf{h}_2, \ldots, \mathbf{h}_N\} \in \mathbb{R}^{N \times 7}$ are customer features, with each node $\mathbf{h}_i = \{[c_i^x, c_i^y], [\delta_i^l, \delta_i^b], [t_i^e, t_i^l, t_i^s]\}$ specifying the coordinates, demands, and time windows. For simplicity, the encoder's input processing and architecture are provided in Appendix B.1.

## 3.3 RELEVANCE-GUIDED CONTEXT REFORMULATION (RGCR)

In multi-constraint scenarios, RGCR automatically learns the relative importance of constraints at each step, enabling the model to focus on the most critical ones. In Fig. 1(b), RGCR undertakes three steps to formulate context embedding: (1) generating embedding for each constraint, (2) computing the correlation between each constraint embedding and the current node embedding, and (3) adaptively aggregating constraint embeddings based on correlation scores.

In the constraint embedding formulation, we first extract the corresponding attributes and then project them through separate linear layers. For the $i$-th trajectory at decoding step $j$, the current node index is denoted as $\tau_{i,j}$. The attributes for each constraint are summarized as follows:

$$
\begin{aligned}
\mathbf{c}_{i,j}^B &= \{\delta_{\tau_{i,j}}^l, \delta_{\tau_{i,j}}^b, c_{i,j}\}, \quad \mathbf{c}_{i,j}^L = \{c_{\tau_{i,j}}^x, c_{\tau_{i,j}}^y, d_{i,j}\}, \\
\mathbf{c}_{i,j}^O &= \{c_{\tau_{i,j}}^x, c_{\tau_{i,j}}^y, d_{i,j}'\}, \quad \mathbf{c}_{i,j}^{TW} = \{t_{\tau_{i,j}}^e, t_{\tau_{i,j}}^l, t_{\tau_{i,j}}^s, t_{i,j}\},
\end{aligned}
\tag{4}
$$

where $\delta^l, \delta^b$ denote the linehaul and backhaul demands, and $c_{i,j}$ is the remaining vehicle capacity. The coordinates $c^x, c^y$ specify node locations in the two-dimensional space, $d$ is the remaining distance of the current sub-route, and $d'$ is the total distance traveled. Moreover, $t^e, t^l, t^s, t$ represent the earliest, latest, service times, and current time, respectively. These attributes are separately fed to linear layers for producing constraint embeddings, denoted as:

$$
\mathbf{C}_{i,j}^k = \mathcal{H}(\mathbf{c}_{i,j}^k),
\tag{5}
$$

where $\mathbf{C}_{i,j}^k \in \mathbb{R}^D$, $k \in \{B, L, O, TW\}$ is the constraint type, and $\mathcal{H}(\cdot)$ denotes the linear layer used for projection. In correlation computing, these constraint embeddings interact with the current node embedding to produce correlation scores, denoted as:

$$
s_{i,j}^k = \mathbf{H}_{\tau_{i,j}} \cdot \mathbf{C}_{i,j}^k,
\tag{6}
$$

where $\mathbf{H}_{\tau_{i,j}} \in \mathbb{R}^D$ is the current node embedding, and $\cdot$ denotes the dot product used for calculating the correlation scores (or similarities). In constraint aggregating, the unified constraint embedding is obtained by adding the original and enhanced constraint embedding, denoted as $\mathbf{S}_{i,j} = \tilde{\mathbf{S}}_{i,j} + \overline{\mathbf{S}}_{i,j}$. The original part is defined as the concatenation of the four constraint embeddings from Eq. (5):

$$
\tilde{\mathbf{S}}_{i,j} = \mathcal{H}(\texttt{Concat}(\mathbf{C}_{i,j}^B, \mathbf{C}_{i,j}^L, \mathbf{C}_{i,j}^O, \mathbf{C}_{i,j}^{TW})),
\tag{7}
$$

where $\texttt{Concat}(\cdot)$ denotes concatenation along the feature dimension, resulting in a concatenated embedding of size $N \times 4D$. $\mathcal{H}(\cdot)$ is a linear layer that projects the $4D$ input back to $D$, resulting in $\tilde{\mathbf{S}}_{i,j} \in \mathbb{R}^D$. For the enhanced part, we apply a weighted sum over the constraint embeddings:

$$
\overline{\mathbf{S}}_{i,j} = s_{i,j}^B \mathbf{C}_{i,j}^B + s_{i,j}^L \mathbf{C}_{i,j}^L + s_{i,j}^O \mathbf{C}_{i,j}^O + s_{i,j}^{TW} \mathbf{C}_{i,j}^{TW}.
\tag{8}
$$

The final context embedding is aggregated from the unified constraint and current node embeddings:

$$
\tilde{\mathbf{C}}_{i,j} = \mathcal{H}(\texttt{Concat}(\mathbf{S}_{i,j}, \mathbf{H}_{\tau_{i,j}})).
\tag{9}
$$

## 3.4 TRAJECTORY-SHARED NODE RE-EMBEDDING (TSNR)

To capture node-specific states influenced by the current context, we aggregate contextual semantics from other nodes and multi-trajectory contexts into the node embeddings. As illustrated in Fig. 1 (c), this is achieved via a multi-head attention mechanism, where node embeddings serve as queries, and the unified node-context information acts as keys and values. Formally, at step $j$, we denote the context embedding for $N$ trajectories as $\tilde{\mathbf{C}}_j = \texttt{Concat}(\tilde{\mathbf{C}}_{1,j}, \tilde{\mathbf{C}}_{2,j}, \ldots, \tilde{\mathbf{C}}_{N,j})$, where $\tilde{\mathbf{C}}_j \in \mathbb{R}^{N \times D}$. By using the last step node $\mathbf{H}_{j-1} \in \mathbb{R}^{(N+1) \times D}$, the query, keys, and values are represented as

$$
\mathbf{q}_j = \mathcal{H}(\texttt{Norm}(\mathbf{H}_{j-1})), \quad \mathbf{k}_j, \mathbf{v}_j = \mathcal{H}(\texttt{Norm}(\texttt{Concat}(\mathbf{H}_{j-1}, \tilde{\mathbf{C}}_j))),
\tag{10}
$$

where $\mathbf{q}_j \in \mathbb{R}^{N \times D}$ and $\mathbf{k}_j, \mathbf{v}_j \in \mathbb{R}^{(N+1) \times D}$. $\texttt{Norm}(\cdot)$ denotes the Root Mean Square (RMS) normalization layer (Zhang and Sennrich, 2019). For simplicity, we use the same notation $\mathcal{H}(\cdot)$ to denote the module that produces $\mathbf{k}_j$ and $\mathbf{v}_j$. To calculate attention weights, we further incorporate a distance-based bias to prevent the model from overfitting to TW. This bias term, denoted as $\mathbf{B}_j = \texttt{Concat}(\mathbf{d}^{n-n}, \mathbf{d}_j^{c-n})$, consists of two parts: the node-node and node-context distance:

$$
\begin{aligned}
\mathbf{d}^{n-n} &= \{d_{m,n} | m, n \in \{0, 1, \ldots, N\}\}, \\
\mathbf{d}_j^{c-n} &= \{d_{m,n} | m \in \{\tau_{1,j}, \tau_{2,j}, \ldots, \tau_{N,j}\}, n \in \{0, 1, \ldots, N\}\},
\end{aligned}
\tag{11}
$$

where $\mathbf{d}^{n-n} \in \mathbb{R}^{(N+1)\times(N+1)}$, $\mathbf{d}_j^{c-n} \in \mathbb{R}^{N\times(N+1)}$ and each element of it takes the form $d_{m,n} = \|\mathbf{c}_m - \mathbf{c}_n\|_2$ with $\mathbf{c} = \{c^x, c^y\}$ denoting Euclidean coordinates. For the node-context part, we extract the coordinates of the current node for each trajectory (indexed by $\{\tau_{1,j}, \tau_{2,j}, \ldots, \tau_{N,j}\}$) and compute their distances to all candidate nodes. The attention weights are subsequently computed as

$$\mathbf{A}_j = \mathtt{Softmax}(\mathbf{q}_j \mathbf{k}_j^\top / \sqrt{D} + \mathbf{B}_j), \tag{12}$$

where $\mathbf{A}_j, \mathbf{B}_j \in \mathbb{R}^{(N+1)\times(N+1+N)}$, and $\mathtt{Softmax}(\cdot)$ is the softmax operation. The re-embedded node representations are computed as follows:

$$\tilde{\mathbf{H}}_j = \mathbf{q}_j + \mathbf{A}_j \mathbf{v}_j, \quad \mathbf{H}_j = \tilde{\mathbf{H}}_j + \mathtt{MLP}(\mathtt{Norm}(\tilde{\mathbf{H}}_j)). \tag{13}$$

We preserve the updated node embeddings $\mathbf{H}_j$ from the current step and use them as input queries for the next step, with update frequency controlled by probabilities $P_{tr}$ (training) and $P_{ts}$ (testing).

### 3.5 Step-wise Decision and Training Objective

Once the context embedding $\tilde{\mathbf{C}}_j \in \mathbb{R}^{N\times D}$ and current node embeddings $\mathbf{H}_j \in \mathbb{R}^{(N+1)\times D}$ are obtained, we use them to predict the selection of the next node, and then compute the RL objective function to optimize model parameters. In the step-wise decision stage, we employ a classic decoder (shown in Appendix B.2) to acquire the probability of selecting the next node. This procedure is represented as follows:

$$\mathbf{P}_j = \mathcal{D}(\tilde{\mathbf{C}}_j, \mathbf{H}_j, \mathbf{M}_j), \tag{14}$$

where $\mathbf{P}_j \in \mathbb{R}^{N\times(N+1)}$, $\mathcal{D}(\cdot)$ denotes the decoder, while $\mathbf{M}_j$ is a mask that prevents revisiting previously selected nodes. If all constraints are satisfied, the node with the highest probability is selected as the next node to visit. Otherwise, the depot is selected. After one interaction, the model generates $N$ solution trajectories, each denoted as $\tau_i = \{\tau_{i,1}, \tau_{i,2}, \ldots, \tau_{i,N'}\}$, where $i \in \{1, 2, \ldots, N\}$ and $N'$ is the total number of decision steps. The RL objective is then computed using the reward of each trajectory and the log-probabilities of selected nodes, as illustrated in Eq. 2.

## 4 Experiments

### 4.1 Datasets and Evaluation Metrics

We evaluate CCL on 48 VRP variants. Following (Berto et al., 2025), node locations are sampled uniformly from the 2D Euclidean space $[0,1)^2$. Each vehicle starts at the depot with a capacity $Q{=}1$ and a maximum route duration $l{=}3$. Linehaul and backhaul demands are sampled as integers from $[1, 10)$ and scaled by a factor of $30 + N/5$, where $N$ is the number of customers. In backhaul settings, 20% of the customers are designated as backhaul, and the remaining 80% as linehaul. For time window tasks, early arrival times, service durations, and time window lengths are independently sampled from $[0.0126, 4.25]$, $[0, 0.15]$, and $[1.8, 2.0)$, respectively. Late times are computed as the sum of early times and window lengths. The training set consists of 100,000 instances uniformly distributed across 16 variants. The best model checkpoint is selected based on validation performance on CVRP (Capacitated VRP), using a held-out set of 128 instances. The test set comprises 48 variants, each containing 1,000 instances. We benchmark CCL against state-of-the-art (SOTA) baselines under two settings: $N{=}50$ and $N{=}100$. We evaluate performance using three standard VRP metrics: total routing length ("Obj."), performance gap ("Gap") to the strong baseline HGS-PyVRP (Wouda et al., 2024), and inference time. All metrics are computed over 1,000 test instances, with "Obj." and "Gap" reported as averages and inference time as total runtime.

### 4.2 Implementation Details

Our method is implemented in PyTorch (Paszke et al., 2019). All experiments are conducted on a machine with an AMD EPYC 7702P 24-core CPU and a single NVIDIA RTX L40S GPU. We use a batch size of 256 during training. The model adopts a 6-layer Transformer encoder, with both encoder and decoder sharing the same architecture: embedding dimension $D{=}128$, 8 attention heads, and a hidden dimension of 512. During decoding, node refinement is applied probabilistically. For instances with $N{=}50$, the refinement probability is 0.75 during training and 1.0

Table 1: Performance on 16 seen in-distribution tasks. * denotes the strong baseline used to compute the gap. Best neural approach is highlighted in **bold**; best existing SOTA is underlined.

| | Methods | $N=50$ Obj.↓ | Gap↓ | Time↓ | $N=100$ Obj.↓ | Gap↓ | Time↓ | | Methods | $N=50$ Obj.↓ | Gap↓ | Time↓ | $N=100$ Obj.↓ | Gap↓ | Time↓ |
|---|---|---|---|---|---|---|---|---|---|---|---|---|---|---|---|
| CVRP | HGS-PyVRP | 10.372 | * | 10.4m | 15.628 | * | 20.8m | VRPTW | HGS-PyVRP | 16.031 | * | 10.4m | 25.423 | * | 20.8m |
| | MTPOMO | 10.520 | 1.423% | 2s | 15.941 | 2.030% | 8s | | MTPOMO | 16.419 | 2.423% | 2s | 26.433 | 3.962% | 9s |
| | MVMoE | 10.499 | 1.229% | 3s | 15.888 | 1.693% | 11s | | MVMoE | 16.400 | 2.298% | 3s | 26.390 | 3.789% | 11s |
| | RF-TE | 10.502 | 1.257% | 2s | 15.860 | 1.524% | 8s | | RF-TE | 16.341 | 1.933% | 2s | 26.228 | 3.154% | 8s |
| | CaDA | 10.505 | 1.287% | 2s | 15.843 | 1.412% | 8s | | CaDA | 16.312 | 1.745% | 1s | 26.169 | 2.925% | 9s |
| | CaDA† | 10.471 | 0.959% | 3s | 15.790 | 1.070% | 13s | | CaDA† | 16.299 | 1.670% | 3s | 26.105 | 2.668% | 14s |
| | CCL | 10.473 | 0.977% | 5s | 15.823 | 1.287% | 19s | | CCL | 16.190 | 0.979% | 5s | 25.913 | 1.908% | 21s |
| | CCL† | **10.463** | **0.881%** | 6s | **15.787** | **1.058%** | 24s | | CCL† | **16.177** | **0.907%** | 7s | 25.862 | 1.706% | 24s |
| OVRP | HGS-PyVRP | 6.507 | * | 10.4m | 9.725 | * | 20.8m | OVRPTW | HGS-PyVRP | 10.510 | * | 10.4m | 16.926 | * | 20.8m |
| | MTPOMO | 6.717 | 3.194% | 2s | 10.216 | 5.028% | 8s | | MTPOMO | 10.676 | 1.558% | 2s | 17.442 | 3.022% | 9s |
| | MVMoE | 6.705 | 3.003% | 3s | 10.177 | 4.617% | 11s | | MVMoE | 10.674 | 1.541% | 3s | 17.416 | 2.870% | 12s |
| | RF-TE | 6.682 | 2.658% | 2s | 10.115 | 3.996% | 8s | | RF-TE | 10.645 | 1.264% | 2s | 17.328 | 2.352% | 9s |
| | CaDA | 6.677 | 2.585% | 1s | 10.095 | 3.786% | 8s | | CaDA | 10.630 | 1.122% | 1s | 17.283 | 2.086% | 9s |
| | CaDA† | 6.652 | 2.212% | 3s | 10.060 | 3.425% | 13s | | CaDA† | 10.621 | 1.030% | 3s | 17.246 | 1.868% | 14s |
| | CCL | 6.636 | 1.957% | 5s | 10.068 | 3.511% | 20s | | CCL | 10.569 | 0.543% | 6s | 17.123 | 1.142% | 21s |
| | CCL† | **6.610** | **1.566%** | 6s | **10.012** | **2.936%** | 25s | | CCL† | **10.564** | **0.506%** | 7s | **17.104** | **1.033%** | 26s |
| OVRPB | HGS-PyVRP | 6.898 | * | 10.4m | 10.335 | * | 20.8m | OVRPBTW | HGS-PyVRP | 11.669 | * | 10.4m | 19.156 | * | 20.8m |
| | MTPOMO | 7.105 | 2.973% | 2s | 10.882 | 5.264% | 8s | | MTPOMO | 11.823 | 1.307% | 3s | 19.656 | 2.592% | 9s |
| | MVMoE | 7.089 | 2.744% | 3s | 10.841 | 4.869% | 11s | | MVMoE | 11.816 | 1.245% | 4s | 19.637 | 2.499% | 13s |
| | RF-TE | 7.065 | 2.385% | 2s | 10.774 | 4.233% | 8s | | RF-TE | 11.790 | 1.027% | 2s | 19.555 | 2.062% | 9s |
| | CaDA | 7.064 | 2.377% | 1s | 10.739 | 3.890% | 8s | | CaDA | 11.775 | 0.898% | 2s | 19.495 | 1.754% | 9s |
| | CaDA† | 7.032 | 1.916% | 3s | 10.682 | 3.329% | 13s | | CaDA† | 11.768 | 0.843% | 3s | 19.469 | 1.617% | 15s |
| | CCL | 7.008 | 1.568% | 5s | 10.666 | 3.179% | 19s | | CCL | 11.721 | 0.436% | 6s | 19.348 | 0.985% | 21s |
| | CCL† | **6.992** | **1.344%** | 6s | **10.624** | **2.775%** | 25s | | CCL† | **11.718** | **0.416%** | 7s | **19.329** | **0.888%** | 27s |
| OVRPBL | HGS-PyVRP | 6.899 | * | 10.4m | 10.335 | * | 20.8m | OVRPBLTW | HGS-PyVRP | 11.668 | * | 10.4m | 19.156 | * | 20.8m |
| | MTPOMO | 7.112 | 3.053% | 2s | 10.888 | 5.318% | 8s | | MTPOMO | 11.823 | 1.315% | 3s | 19.658 | 2.602% | 9s |
| | MVMoE | 7.094 | 2.799% | 3s | 10.847 | 4.929% | 11s | | MVMoE | 11.816 | 1.249% | 4s | 19.640 | 2.514% | 12s |
| | RF-TE | 7.068 | 2.417% | 2s | 10.778 | 4.266% | 8s | | RF-TE | 11.789 | 1.017% | 2s | 19.554 | 2.061% | 9s |
| | CaDA | 7.062 | 2.339% | 1s | 10.741 | 3.900% | 8s | | CaDA | 11.777 | 0.914% | 2s | 19.497 | 1.762% | 9s |
| | CaDA† | 7.034 | 1.935% | 3s | 10.686 | 3.368% | 13s | | CaDA† | 11.769 | 0.848% | 3s | 19.467 | 1.602% | 15s |
| | CCL | 7.009 | 1.569% | 5s | 10.681 | 3.323% | 20s | | CCL | 11.721 | 0.442% | 6s | 19.346 | 0.977% | 22s |
| | CCL† | **6.992** | **1.335%** | 6s | **10.609** | **2.631%** | 23s | | CCL† | **11.718** | **0.414%** | 7s | **19.334** | **0.915%** | 27s |
| OVRPL | HGS-PyVRP | 6.507 | * | 10.4m | 9.724 | * | 20.8m | OVRPLTW | HGS-PyVRP | 10.510 | * | 10.4m | 16.926 | * | 20.8m |
| | MTPOMO | 6.720 | 3.248% | 2s | 10.224 | 5.112% | 8s | | MTPOMO | 10.677 | 1.572% | 2s | 17.442 | 3.020% | 9s |
| | MVMoE | 6.706 | 3.028% | 3s | 10.184 | 4.693% | 11s | | MVMoE | 10.677 | 1.564% | 3s | 17.418 | 2.880% | 12s |
| | RF-TE | 6.683 | 2.680% | 2s | 10.121 | 4.054% | 8s | | RF-TE | 10.646 | 1.267% | 2s | 17.328 | 2.352% | 9s |
| | CaDA | 6.680 | 2.623% | 1s | 10.093 | 3.773% | 8s | | CaDA | 10.631 | 1.133% | 1s | 17.280 | 2.073% | 9s |
| | CaDA† | 6.652 | 2.200% | 2s | 10.060 | 3.432% | 13s | | CaDA† | 10.621 | 1.033% | 3s | 17.244 | 1.861% | 14s |
| | CCL | 6.637 | 1.968% | 5s | 10.067 | 3.495% | 20s | | CCL | 10.569 | 0.546% | 6s | 17.123 | 1.143% | 22s |
| | CCL† | **6.610** | **1.569%** | 6s | **10.000** | **2.811%** | 24s | | CCL† | **10.564** | **0.501%** | 7s | **17.109** | **1.063%** | 26s |
| VRPB | HGS-PyVRP | 9.687 | * | 10.4m | 14.377 | * | 20.8m | VRPBTW | HGS-PyVRP | 18.292 | * | 10.4m | 29.467 | * | 20.8m |
| | MTPOMO | 10.036 | 3.596% | 2s | 15.102 | 5.052% | 8s | | MTPOMO | 18.649 | 1.938% | 2s | 30.478 | 3.426% | 9s |
| | MVMoE | 10.007 | 3.292% | 3s | 15.023 | 4.505% | 10s | | MVMoE | 18.632 | 1.841% | 3s | 30.437 | 3.284% | 12s |
| | RF-TE | 9.979 | 3.000% | 2s | 14.935 | 3.906% | 8s | | RF-TE | 18.573 | 1.517% | 2s | 30.249 | 2.641% | 9s |
| | CaDA | 9.979 | 3.010% | 1s | 14.910 | 3.721% | 8s | | CaDA | 18.543 | 1.361% | 1s | 30.174 | 2.390% | 9s |
| | CaDA† | 9.922 | 2.405% | 2s | 14.838 | 3.222% | 13s | | CaDA† | 18.528 | 1.276% | 3s | 30.113 | 2.183% | 14s |
| | CCL | 9.916 | 2.352% | 5s | 14.882 | 3.526% | 19s | | CCL | 18.430 | 0.738% | 6s | 29.911 | 1.494% | 21s |
| | CCL† | **9.875** | **1.921%** | 6s | **14.780** | **2.808%** | 22s | | CCL† | **18.419** | **0.678%** | 7s | **29.871** | **1.357%** | 26s |
| VRPBL | HGS-PyVRP | 10.186 | * | 10.4m | 14.779 | * | 20.8m | VRPBLTW | HGS-PyVRP | 18.361 | * | 10.4m | 29.026 | * | 20.8m |
| | MTPOMO | 10.679 | 4.760% | 2s | 15.718 | 6.294% | 8s | | MTPOMO | 19.001 | 2.199% | 3s | 30.948 | 3.794% | 9s |
| | MVMoE | 10.639 | 4.384% | 3s | 15.642 | 5.771% | 11s | | MVMoE | 18.983 | 2.097% | 3s | 30.892 | 3.609% | 12s |
| | RF-TE | 10.569 | 3.713% | 2s | 15.523 | 5.008% | 8s | | RF-TE | 18.910 | 1.713% | 2s | 30.705 | 2.978% | 9s |
| | CaDA | 10.576 | 3.776% | 1s | 15.490 | 4.771% | 8s | | CaDA | 18.894 | 1.623% | 1s | 30.620 | 2.700% | 9s |
| | CaDA† | 10.503 | 3.064% | 3s | 15.389 | 4.093% | 13s | | CaDA† | 18.878 | 1.540% | 3s | 30.570 | 2.531% | 15s |
| | CCL | 10.484 | 2.883% | 5s | 15.407 | 4.219% | 19s | | CCL | 18.773 | 0.976% | 6s | 30.366 | 1.842% | 21s |
| | CCL† | **10.440** | **2.450%** | 6s | **15.297** | **3.472%** | 24s | | CCL† | **18.758** | **0.899%** | 7s | **30.323** | **1.697%** | 25s |
| VRPL | HGS-PyVRP | 10.587 | * | 10.4m | 15.766 | * | 20.8m | VRPLTW | HGS-PyVRP | 16.356 | * | 10.4m | 25.757 | * | 20.8m |
| | MTPOMO | 10.775 | 1.733% | 2s | 16.157 | 2.483% | 8s | | MTPOMO | 16.832 | 2.877% | 2s | 26.913 | 4.455% | 9s |
| | MVMoE | 10.753 | 1.525% | 3s | 16.099 | 2.113% | 11s | | MVMoE | 16.817 | 2.783% | 3s | 26.866 | 4.272% | 12s |
| | RF-TE | 10.747 | 1.485% | 2s | 16.057 | 1.858% | 8s | | RF-TE | 16.728 | 2.248% | 2s | 26.706 | 3.645% | 9s |
| | CaDA | 10.749 | 1.505% | 1s | 16.036 | 1.725% | 8s | | CaDA | 16.709 | 2.130% | 1s | 26.631 | 3.358% | 9s |
| | CaDA† | 10.707 | 1.112% | 2s | 15.984 | 1.400% | 13s | | CaDA† | 16.692 | 2.034% | 3s | 26.556 | 3.065% | 14s |
| | CCL | 10.710 | 1.145% | 5s | 16.009 | 1.561% | 19s | | CCL | 16.579 | 1.333% | 6s | 26.366 | 2.321% | 20s |
| | CCL† | **10.698** | **1.027%** | 6s | **15.960** | **1.245%** | 23s | | CCL† | **16.556** | **1.192%** | 7s | **26.324** | **2.157%** | 24s |

during testing. For $N$=100, the respective probabilities are 0.25 and 0.5. The model is optimized using Adam with a learning rate of $3 \times 10^{-4}$ and a weight decay of $1 \times 10^{-6}$. A multi-step learning rate scheduler is used with milestones at epochs 270 and 295, a decay factor of 0.1, and gradient clipping set to 1. Training is conducted for a total of 300 epochs. **Our code is available at `https://github.com/gshuangchun/CCL-MTLVRP.git`**.

## 4.3 COMPARISON WITH THE STATE-OF-THE-ARTS

**Baselines.** We compare CCL with state-of-the-art multi-task VRP solvers, including MTPOMO (Liu et al., 2024), MVMoE (Zhou et al., 2024a), RouteFinder (RF-TE) (Berto et al., 2025), and CaDA (Li et al., 2025). Among these, RF-TE and CaDA have reported the strongest performance, and we

Table 2: Generalization on 32 unseen out-of-distribution tasks.

| Methods | MDOVRPB, MDOVRPL, MDVRPBL, MDOVRPBL MDCVRP, MDOVRP, MDVRPB, MDVRPL VRPMB, OVRPMB, VRPMBL, OVRPMBL MDVRPMB, MDOVRPMB, MDVRPMBL, MDOVRPMBL | | | | | | MDOVRPBTW, MDOVRPLTW, MDVRPBLTW, MDOVRPBLTW MDCVRPTW, MDOVRPTW, MDVRPBTW, MDVRPLTW VRPMBTW, OVRPMBTW, VRPMBLTW, OVRPMBLTW MDVRPMBTW, MDOVRPMBTW, MDVRPMBLTW, MDOVRPMBLTW | | | | | |
|---|---|---|---|---|---|---|---|---|---|---|---|---|
| | $N=50$ | | | $N=100$ | | | $N=50$ | | | $N=100$ | | |
| | Obj. ↓ | Gap ↓ | Time ↓ | Obj. ↓ | Gap ↓ | Time ↓ | Obj. ↓ | Gap ↓ | Time ↓ | Obj. ↓ | Gap ↓ | Time ↓ |
| RF-TE | 9.651 | 40.472% | 1s | 14.746 | 45.724% | 9s | 14.557 | 32.586% | 2s | 24.217 | 36.564% | 10s |
| CaDA | 9.535 | 38.860% | 2s | 14.910 | 47.156% | 10s | 14.410 | 30.872% | 2s | 23.523 | 32.512% | 10s |
| CaDA† | 9.285 | 34.169% | 3s | 14.395 | 41.419% | 16s | 14.830 | 34.665% | 4s | 24.328 | 36.774% | 17s |
| CCL | 8.906 | 29.156% | 4s | 14.071 | 38.624% | 17s | 13.923 | 26.020% | 4s | **23.375** | **31.159%** | 18s |
| CCL† | **8.673** | **25.781%** | 5s | **13.777** | **35.413%** | 22s | **13.536** | **22.422%** | 5s | 24.025 | 34.163% | 25s |

Table 3: Ablation on key modules within CCL.

| Methods | CVRP | OVRP | VRPB | VRPL | OVRPB | OVRPL | VRPBL | OVRPBL | Avg. |
|---|---|---|---|---|---|---|---|---|---|
| CCL† | 0.881% | **1.566%** | **1.921%** | 1.027% | **1.344%** | **1.569%** | **2.450%** | **1.335%** | **1.512%** |
| - RGCR | **0.874%** | 1.710% | 1.969% | **0.993%** | 1.396% | 1.712% | 2.486% | 1.407% | 1.568% |
| - TSNR | 0.961% | 2.284% | 2.413% | 1.131% | 1.969% | 2.311% | 3.001% | 1.973% | 2.005% |
| - RGCR - TSNR | 1.014% | 2.395% | 2.416% | 1.160% | 2.036% | 2.411% | 3.088% | 2.041% | 2.070% |

| Methods | VRPTW | OVRPTW | VRPBTW | VRPLTW | OVRPBTW | OVRPLTW | VRPBLTW | OVRPBLTW | Avg. |
|---|---|---|---|---|---|---|---|---|---|
| CCL† | **0.907%** | **0.506%** | **0.678%** | **1.192%** | **0.416%** | **0.501%** | **0.899%** | **0.414%** | **0.689%** |
| - RGCR | 0.938% | 0.521% | 0.720% | 1.235% | 0.419% | 0.519% | 0.926% | 0.427% | 0.713% |
| - TSNR | 1.539% | 0.947% | 1.204% | 1.857% | 0.795% | 0.957% | 1.409% | 0.805% | 1.189% |
| - RGCR - TSNR | 1.615% | 0.969% | 1.266% | 1.930% | 0.813% | 0.958% | 1.492% | 0.810% | 1.232% |

include them in both in-distribution (Table 1) and out-of-distribution (Table 2) evaluations. To ensure a fair comparison, we reimplement CaDA in the RouteFinder framework (Berto et al., 2025), which also serves as the basis for RF-TE and our CCL. As shown in Appendix C.1.1, our reproduction closely matches the performance reported in the original paper (Li et al., 2025). To further enhance performance, we integrate a context-aware module, ReLD (Huang et al., 2025), into both CaDA and CCL, denoted as CaDA† and CCL†, respectively.

**In-Distribution Evaluation.** In the Table 1, CCL outperforms CaDA across both $N$=50 and $N$=100 settings. Specifically, for $N$=50, both CCL and CCL† achieve lower performance gaps than CaDA on all 16 evaluated tasks. For $N$=100, the gap relative to the HGS-PyVRP baseline narrows even further. We also observe complementary strengths between ReLD and CCL. ReLD performs particularly well on variants without time windows (TW), leveraging its ability to extract globally shared constraint signals through step-wise context. In contrast, CCL's dynamic node refinement excels on TW tasks, offering finer-grained adaptation to local, node-specific constraints. By combining both, the resulting CCL† achieves the best overall performance across all the in-distribution tasks.

**Out-of-Distribution Evaluation.** We present the averaged performance for both tasks with or without TW in Table 2 (detailed results for each task are presented in Appendix C.1.3, where CCL outperformed CaDA on the majority). Table 2 shows that CCL† consistently outperforms other methods under the $N$=50 setting. For $N$=100, while CCL† maintains competitive performance, it shows a slightly higher gap on TW tasks compared to standalone CCL. We hypothesize that this may be due to a low test-time update rate, which can cause the model to overfit to static constraint structures and under-adapt to time-sensitive variations, thus increasing the gap. Nevertheless, either equipped with ReLD or not, our CCL exhibits superior overall performance to CaDA (and its counterpart).

## 4.4 ABLATION STUDIES

**Ablation on Key Modules within CCL.** We conduct ablation experiments to validate the effectiveness of RGCR and TSNR in CCL†. Table 3 reports the results for $N$=50. Removing RGCR leads to a smaller gap increase than TSNR, and even slightly reduces the gap on CVRP and VRPL. It is likely that the relatively simple constraints of the two VRP tasks make relevance weighting less effective. Moreover, removing both modules yields the highest gap, highlighting their complementary effectiveness. We also conduct these ablations on CCL (the variant without ReLD), showing that the main performance gains come from CCL itself rather than ReLD. Details are presented in Appendix C.3.1.

| Methods | Gap (%) | | | Complexity | |
| | Avg. | w/o TW | w/ TW | # Params (M) | Time (s) |
|---|---|---|---|---|---|
| Concat Attributes | 1.156 | 1.584 | 0.729 | **3.95** | **5.2** |
| Concat Embeddings | 1.141 | 1.568 | 0.713 | 4.05 | 5.7 |
| + Random Scores | **1.092** | **1.477** | 0.707 | 4.05 | 6.5 |
| + Cosine Similarity | 1.108 | 1.480 | 0.737 | 4.05 | 7.2 |
| + Dot Product (RGCR) | 1.100 | 1.512 | **0.689** | 4.05 | 6.5 |

Figure 2: Ablation on key components within RGCR (**Left**) and TSNR (**Right**), respectively.

**Ablation on Key Components within RGCR and TSNR.** We further conduct ablation studies using $N$=50 to evaluate the components within RGCR and TSNR.

Regarding RGCR, we first examine direct concatenation of attributes (as in CaDA) and embeddings (CCL$^\dagger$-RGCR in Table 3). We then evaluate three correlation scores, namely random, cosine similarity, and dot product, as defined in Eq. 6. The left part of Fig. 2 presents the averaged gap and the corresponding model complexity. Compared with direct concatenation, all three correlation scores reduce the gap in both settings. Notably, the dot product achieves the smallest gap on tasks with TW, demonstrating its superiority in handling complex constraints. More experimental setups and analyses are provided in Appendix C.3.2.

Regarding TSNR, the right part of Fig. 2 shows that combining node-level attention, embedding updates, and the distance bias in Eq. 12 achieves the lowest gaps, indicating that all three elements are essential for improving the overall performance. Moreover, a detailed analysis that node-level attention reduces model complexity compared to the vanilla Transformer is provided in Appendix C.3.3.

## 5 DISCUSSION

We discuss the strengths and limitations of CCL. Its main drawback is the longer inference time required for better performance. However, flexible parameter settings can mitigate this issue, enabling CCL to perform well on large-scale real-world instances.

**Complexity Analysis.** Table 4 compares the model complexity of SOTA methods and our CCL. We further compare CCL with a heavy-decoder variant of the SOTA model, denoted as CaDA$^\dagger$-HD. Detailed configurations of this variant are presented in Appendix C.4. All models are trained and evaluated on the 16 VRP variants with $N$=50 using an L40S GPU. "Time" denotes the total inference time over 1,000 test instances, "# Params" refers to the

Table 4: Complexity analysis.

| Methods | Gap↓ (%) | Memory↓ (GiB) | # Params↓ (M) | Time↓ (s) |
|---|---|---|---|---|
| CaDA | 1.90 | 6.01 | 3.37 + 0.1 | 1.5 |
| CaDA$^\dagger$ | 1.63 | 7.15 | 3.37 + 0.3 | 2.7 |
| CaDA$^\dagger$-HD | 1.53 | 7.55 | 3.37+1.0 | 5.1 |
| CCL | 1.28 | 8.13 | 3.39 + 0.45 | 5.4 |
| CCL$^\dagger$ | 1.10 | 8.76 | 3.39 + 0.66 | 6.5 |
| ASW-TAM | 29.93 | 15.47 | 3.39 + 0.66 | 96.1 |

total number of parameters in the encoder and decoder, and "Memory" indicates the peak memory usage during testing across all 16 variants. Compared to CaDA and CaDA$^\dagger$, our CCL and CCL$^\dagger$ introduce only a moderate increase in memory usage and parameter count, while achieving a substantial performance improvement. The inference time is longer due to additional computation, but the gain in solution quality justifies the cost. In addition, we also apply the step-wise refinement strategy, *i.e.*, ASW-TAM (Xin et al., 2020) to the multi-task setting, where each route is re-embedded individually. However, due to memory constraints, we adopt a much smaller batch size that is only 1/16 of the original one. Results show that the naive refinement strategy leads to significantly higher gaps, longer inference time, and larger memory consumption, which further validates the effectiveness of CCL. Moreover, we observe that CCL achieves a comparable model cost while reducing the gap by 0.25% compared with CaDA$^\dagger$-HD. These findings indicate that the effectiveness of CCL stems from its design rather than from an increased network scale.

**Performance-Cost Trade-off.** In Section 3.4, $P_{tr}$ and $P_{ts}$ denote the probabilities of updating node embeddings during training and testing, and we assess their impact on model performance and inference efficiency, which also leads to a lightweight version of CCL. We first conduct sensitivity studies using $N$=50 and evaluating all 20 combinations of $P_{tr} \in \{0.25, 0.5, 0.75, 1\}$ and $P_{ts} \in \{0, 0.25, 0.5, 0.75, 1\}$. Fig. 3 shows the corresponding gap values and inference time.

We observe that, for a fixed probability $P_{tr}$ during training, the gap tends to be smaller when the probability $P_{ts}$ during testing is slightly higher. For example, when $P_{tr}$=0.25 or 0.5, the best performance is achieved at $P_{ts}$=0.5 and 0.75, respectively. This work adopts $P_{tr} = 0.75$ and $P_{ts} = 1$, as this setting yields the lowest gap. Meanwhile, reducing $P_{ts}$ leads to shorter inference time, as fewer re-

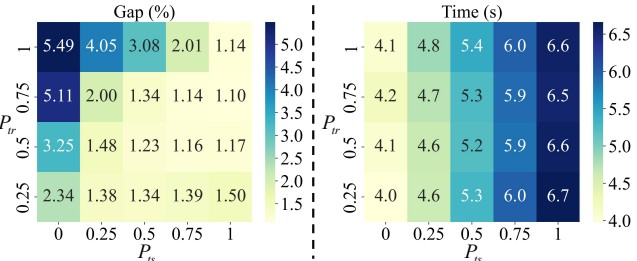

Figure 3: Gap (**Left**) and inference time (**Right**) under different training and testing update rates.

finement steps are involved. While this leads to a higher gap, it provides a trade-off between solution quality and inference efficiency. Motivated by this, we design a lightweight version of CCL[†] using $P_{tr} = 0.25$ and $P_{ts} = 0.25$. It achieves an average gap of 1.38% across 16 VRPs with an average inference time of 4.6s, while the existing SOTA CaDA[†] attains a gap of 1.63% in 2.7s (see Table 4). This enables users to adjust $P_{ts}$ based on the requirements of practical deployment scenarios, and more detailed comparisons between CCL and CaDA are provided in Appendix C.4.

**Large-Scale Real-World Practicality.** We evaluate zero-shot generalization on 60 real-world VRPTW instances, each with $N$=600 (Homberger and Gehring, 1999). The model is trained on 16 tasks with $N$=100. During inference, we apply an update probability of $P_{ts}$=0.1 to reduce computational cost. Table 5 reports the averaged results across these instances (per-instance results in Appendix C.7), showing that CCL achieves the lowest average gap while maintaining comparable infer-

Table 5: Results on large-scale real-world VRPTW instances (N=600).

| Methods | Obj.↓ | Gap↓ | Time↓ |
|---|---|---|---|
| RF-TE | 29558 | 145.593% | **1.4s** |
| CaDA | 22917 | 88.188% | **1.4s** |
| CCL | **20633** | **70.961%** | 2.0s |

ence time. Appendix C.7 also reports results on VRPTW instances with $N$=100 (Solomon, 1987), where CCL outperforms SOTAs on 24 out of 27. It is further evaluated on CVRP instances with $N \in [100, 251]$ (Uchoa et al., 2017), where CCL achieves the best performance on 16 out of 27. These findings indicate that our method is well-suited for deployment in real-world scenarios, particularly for problems with complex constraints such as time windows.

## 6    CONCLUSIONS

Existing neural multi-task VRP methods often neglect the evolving nature of node states during decoding, limiting their ability to respond accurately to constraint requirements. To overcome this, we proposed Chain-of-Context Learning (CCL), a step-wise framework that updates node embeddings based on the current decision context. Through relevance-guided constraint reformulation and trajectory-shared re-embedding, CCL captures the agent's evolving preferences and improves solution quality. Experiments on 48 VRP variants show that CCL achieves SOTA performances on all in-distribution and most out-of-distribution tasks. One limitation of CCL lies in its slightly longer inference time. Although flexible parameter settings can mitigate such issue, a trade-off between computation cost and solution quality still remains. In future, we plan to explore more advanced techniques to further improve the inference efficiency while preserving superior solution quality.

### THE USE OF LARGE LANGUAGE MODELS (LLMS)

We employed LLMs for polishing the paper and assisting with simple coding tasks.

### ACKNOWLEDGMENTS

This research is supported by the National Research Foundation, Singapore under its AI Singapore Programme (AISG Award No: AISG3-RP-2022-031). This research is also supported by the Lee Kong Chian Fellowship awarded to CAO Zhiguang by Singapore Management University.

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

## A  RELATED WORK

**Neural Solvers for Single-Task VRPs.** A common paradigm in neural solvers for single-task VRPs is to construct solutions in an autoregressive manner. These methods typically employ an encoder to embed the VRP instance into node representations, followed by a decoder that sequentially predicts the probability of selecting the next node. To reduce the computational overhead of reinforcement learning (RL), most approaches adopt *static node embeddings* during decoding (Joshi et al., 2019; Nazari et al., 2018; Kool et al., 2019; Kwon et al., 2020; Huang et al., 2025; Li et al., 2021a; Hou et al., 2023; Ye et al., 2024; Joshi et al., 2021; Bi et al., 2022; Geisler et al., 2022; Zhou et al., 2023; Zhang and Cao, 2025; Zhang et al., 2025). One influential method in this line is the Attention Model (AM) (Kool et al., 2019), which uses a Transformer-based policy network to guide node selection. (Kwon et al., 2020) enhances AM by introducing Policy Optimization with Multiple Optima (POMO), which leverages multiple solution trajectories and data augmentation to achieve strong performance on TSP and CVRP. POMO has since become a widely adopted baseline (Kwon et al., 2021; Li et al., 2021b; Kim et al., 2022; Grinsztajn et al., 2023; Chen et al., 2023b; Gao et al., 2024; Hottung et al., 2025; Hua et al., 2025a; Li et al., 2021a; Hou et al., 2023; Ye et al., 2024; Joshi et al., 2021; Bi et al., 2022; Geisler et al., 2022; Zhou et al., 2023). To improve context modeling, ReLD (Huang et al., 2025) proposes an enhanced decoder architecture incorporating identity mapping and a feed-forward layer to better capture local and global dependencies. In terms of *dynamic node re-embedding*, (Xin et al., 2020) introduces a step-wise RL framework that removes visited nodes at each decision step, enabling the model to represent distinct node states as the context evolves. An alternative direction involves building heavy decoder-based solvers trained with supervised learning (SL) (Drakulic et al., 2023; Luo et al., 2023; 2025; Pirnay and Grimm, 2024; Drakulic et al., 2025). While these methods demonstrate strong empirical performance, their reliance on multi-layered decoder architectures results in high computational cost, making them unsuitable for RL-based training, which does not need optimal solutions as the labels.

**Neural Solvers for Multi-Task VRPs.** Multi-task VRPs generalize single-task VRPs by involving varied combinations of constraints, resulting in multiple task variants within a shared framework. Recent works train a single model to capture transferable patterns across tasks (Lin et al., 2024; Liu et al., 2024; Zhou et al., 2024a; Berto et al., 2024; Zong et al., 2025; Drakulic et al., 2025; Jiang et al., 2024; Zhou et al., 2024b; Hua et al., 2025b;c; Li et al., 2025; Berto et al., 2025; Son et al., 2025; Wang et al., 2025; Liu et al., 2026; 2025). (Lin et al., 2024) shows that a pre-trained TSP model can be fine-tuned to handle other VRP variants. To expand constraint coverage, (Liu et al., 2024) and (Zhou et al., 2024a) introduce models that handle B, L, O, and TW constraints, training on single-constraint tasks with the goal of generalizing to tasks with mixed attributes. (Berto et al., 2024) presents a foundation model trained on 16 variants and fine-tuned on an unseen constraint (MB) across 8 new variants. A subsequent extension (Berto et al., 2025) adds MD to the task space, culminating in a benchmark of 48 variants. Most recently, (Li et al., 2025) proposes Constraint-Aware Dual-Attention (CaDA), which incorporates constraint prompts and global-sparse attention to enhance encoder performance in capturing both broad and localized constraint-relevant node information. Despite their progress, these methods generally struggle to capture the dynamic, fine-grained impact of constraints during decision-making—particularly when certain nodes become increasingly urgent due to time-sensitive or context-dependent requirements. In contrast, our work introduces a step-wise, context-aware refinement mechanism to better model these evolving constraint-driven priorities.

## B  DETAILS OF MODEL ARCHITECTURE

### B.1  ENCODER ARCHITECTURE

As illustrated in Section 4.2, the input attributes includes three parts: the constraint label $\tilde{\mathbf{h}}_0 \in \mathbb{R}^4$, the depot attribute $\mathbf{h}_0 \in \mathbb{R}^4$, and the customer features $\{\mathbf{h}_1, \mathbf{h}_2, \ldots, \mathbf{h}_N\} \in \mathbb{R}^{N \times 7}$. The depot and customer attributes are separately processed by two linear layers, and the outputs are concatenated to generate the input node embedding as follows:

$$\mathbf{I} = \texttt{Concat}(\mathcal{H}(\mathbf{h}_0), \mathcal{H}(\{\mathbf{h}_1, \mathbf{h}_2, \ldots, \mathbf{h}_N\})), \tag{15}$$

where $\mathbf{I} \in \mathbb{R}^{(N+1) \times D}$, $\texttt{Concat}(\cdot)$ denotes the concatenate operation, and $\mathcal{H}(\cdot)$ represents the linear layer. Similarly, we project the constraint labels into the prompt embedding space and expand them

to match the shape of the node embeddings:

$$\mathbf{L} = \texttt{Expand}(\mathcal{H}(\tilde{\mathbf{h}}_0)), \tag{16}$$

where $\mathbf{L} \in \mathbb{R}^{(N+1) \times D}$, $\mathcal{H}(\cdot)$ is a linear layer to project the feature dimension from 4 to $D$, and $\texttt{Expand}(\cdot)$ is the duplication and expansion operation to reshape the feature map from $\mathbb{R}^{1 \times D}$ to $\mathbb{R}^{(N+1) \times D}$. Subsequently, a unified input embedding is formed by concatenating the projected constraint embedding and the node embedding:

$$\tilde{\mathbf{I}} = \mathcal{H}(\texttt{Concat}(\mathbf{I}, \mathbf{L})). \tag{17}$$

The dual-attention encoder processes the original node embeddings $\mathbf{I}$ and the unified input embeddings $\tilde{\mathbf{I}}$ through the sparse and global branches, respectively. Each branch contains a Transformer layer and a linear layer for fusion, with the overall computation defined as:

$$\begin{aligned}
\tilde{\mathbf{H}}^{'(i)} &= \mathcal{T}_g^{(i)}(\tilde{\mathbf{H}}^{(i-1)}), \quad \mathbf{H}^{'(i)} = \mathcal{T}_s^{(i)}(\mathbf{H}^{(i-1)}), \\
\tilde{\mathbf{H}}^{(i)} &= \tilde{\mathbf{H}}^{'(i)} + \mathcal{H}_g^{(i)}(\mathbf{H}^{'(i)}), \quad \mathbf{H}^{(i)} = \mathbf{H}^{'(i)} + \mathcal{H}_s^{(i)}(\tilde{\mathbf{H}}^{'(i)}),
\end{aligned} \tag{18}$$

where $\tilde{\mathbf{H}}^{'}, \tilde{\mathbf{H}}, \mathbf{H}^{'}, \mathbf{H} \in \mathbb{R}^{(N+1) \times D}$, and $i$ denotes the index of the encoder layer. In the first layer, we initialize $\tilde{\mathbf{H}}^{(1)} = \tilde{\mathbf{I}}$ and $\mathbf{H}^{(1)} = \mathbf{I}$. $\mathcal{T}_g^{(i)}, \mathcal{H}_g^{(i)}$ denote the Transformer and linear layers of the global branch, respectively, while $\mathcal{T}_s^{(i)}, \mathcal{H}_s^{(i)}$ correspond to those of the sparse branch. The Transformer layer employs the pre-norm design from (Berto et al., 2024), and integrates sparse attention based on (Li et al., 2025). The embedding output by the final global layer is passed through a normalization layer, and the normalized embeddings are used as the initial node embeddings for decoding:

$$\mathbf{H} = \texttt{Norm}(\mathbf{H}^{(K)}), \tag{19}$$

where $\mathbf{H} \in \mathbb{R}^{(N+1) \times D}$ and $K = 6$ denotes the number of encoder layers.

## B.2 CLASSIC DECODER

Following RGCR and TSNR, a classic decoder is employed to calculate the action probability distribution using a multi-head attention mechanism. At step $j$, the context embedding generated by RGCR is denoted as $\tilde{\mathbf{C}}_j \in \mathbb{R}^{N \times D}$, where $N$ is the number of trajectories, equal to the number of customers. We directly use the context embeddings as the query, *i.e.*, $\mathbf{q}_j = \tilde{\mathbf{C}}_j$. In the multi-head attention mechanism, the key and value embeddings are derived from the node embeddings produced by TSNR, *i.e.*, $\mathbf{k}_j, \mathbf{v}_j = \mathcal{H}(\mathbf{H}_j)$, where $\mathbf{k}_j, \mathbf{v}_j \in \mathbb{R}^{(N+1) \times D}$. Since each node is visited only once, we apply a mask $\mathbf{M}_j$ to the visited nodes when computing the attention weights:

$$\mathbf{A}_j = \texttt{Softmax}(\frac{\mathbf{q}_j \mathbf{k}_j^\top}{\sqrt{D}} \odot \mathbf{M}_j), \tag{20}$$

where $\mathbf{A}_j, \mathbf{M}_j \in \mathbb{R}^{N \times (N+1)}$, $\texttt{Softmax}(\cdot)$ denotes the Softmax operation, and $\odot$ ensures that multiplication values for visited nodes are set to $-\infty$. The context query is computed as $\tilde{\mathbf{q}}_j = \mathcal{H}(\mathbf{A}_j \mathbf{v}_j)$, and the candidate node representations are obtained as $\tilde{\mathbf{k}}_j = \mathcal{H}(\mathbf{H}_j)$. Based on these, the action probability distribution is derived as follows:

$$\mathbf{D}_j = \frac{\tilde{\mathbf{q}}_j \tilde{\mathbf{k}}_j^\top}{\sqrt{D}}, \tag{21}$$

where $\tilde{\mathbf{q}}_j \in \mathbb{R}^{N \times D}$, $\tilde{\mathbf{k}}_j \in \mathbb{R}^{(N+1) \times D}$, and $\mathbf{D}_j \in \mathbb{R}^{N \times N+1}$. To generate solutions, the unnormalized log-probability (logit) is calculated as

$$\mathbf{u}_j = \xi \cdot \texttt{Tanh}(\mathbf{D}_j) \odot \mathbf{M}_j, \tag{22}$$

where $\texttt{Tanh}(\cdot)$ is the Hyperbolic Tangent operation, and $\xi = 10$ is a predefined clipping hyperparameter. The final selection probabilities for each node are computed by applying the Softmax operation: $\mathbf{P}_j = \texttt{Softmax}(\mathbf{u}_j)$.

Table 5: Performance on 16 seen in-distribution tasks. * denotes the strong baseline used to compute the gap. Best neural approach is highlighted in **bold**; second underlined.

| | Methods | N=50 Obj. ↓ | Gap ↓ | Time ↓ | N=100 Obj. ↓ | Gap ↓ | Time ↓ | | Methods | N=50 Obj. ↓ | Gap ↓ | Time ↓ | N=100 Obj. ↓ | Gap ↓ | Time ↓ |
|---|---|---|---|---|---|---|---|---|---|---|---|---|---|---|---|
| CVRP | HGS-PyVRP | 10.372 | * | 10.4m | 15.628 | * | 20.8m | VRPTW | HGS-PyVRP | 16.031 | * | 10.4m | 25.423 | * | 20.8m |
| | CaDA‡ | 10.494 | 1.182% | 2s | 15.870 | 1.578% | 8s | | CaDA‡ | 16.278 | 1.536% | 2s | 26.070 | 2.530% | 8s |
| | CaDA | 10.505 | 1.287% | 2s | 15.843 | 1.412% | 8s | | CaDA | 16.312 | 1.745% | 1s | 26.169 | 2.925% | 9s |
| | CCL | **10.473** | **0.977%** | 5s | **15.823** | **1.287%** | 19s | | CCL | **16.190** | **0.979%** | 5s | **25.913** | **1.908%** | 21s |
| OVRP | HGS-PyVRP | 6.507 | * | 10.4m | 9.725 | * | 20.8m | OVRPTW | HGS-PyVRP | 10.510 | * | 10.4m | 16.926 | * | 20.8m |
| | CaDA‡ | 6.670 | 2.468% | 2s | 10.121 | 4.045% | 8s | | CaDA‡ | 10.613 | 0.957% | 2s | 17.226 | 1.751% | 9s |
| | CaDA | 6.677 | 2.585% | 1s | 10.095 | 3.786% | 8s | | CaDA | 10.630 | 1.122% | 1s | 17.283 | 2.086% | 9s |
| | CCL | **6.636** | **1.957%** | 5s | **10.068** | **3.511%** | 20s | | CCL | **10.569** | **0.543%** | 6s | **17.123** | **1.142%** | 21s |
| OVRPB | HGS-PyVRP | 6.898 | * | 10.4m | 10.335 | * | 20.8m | OVRPBTW | HGS-PyVRP | 11.669 | * | 10.4m | 19.156 | * | 20.8m |
| | CaDA‡ | 7.049 | 2.159% | 2s | 10.762 | 4.099% | 8s | | CaDA‡ | 11.761 | 0.779% | 2s | 19.436 | 1.441% | 9s |
| | CaDA | 7.064 | 2.377% | 1s | 10.739 | 3.890% | 8s | | CaDA | 11.775 | 0.898% | 2s | 19.495 | 1.754% | 9s |
| | CCL | **7.008** | **1.568%** | 5s | **10.666** | **3.179%** | 19s | | CCL | **11.721** | **0.436%** | 6s | **19.348** | **0.985%** | 21s |
| OVRPBL | HGS-PyVRP | 6.899 | * | 10.4m | 10.335 | * | 20.8m | OVRPBLTW | HGS-PyVRP | 11.668 | * | 10.4m | 19.156 | * | 20.8m |
| | CaDA‡ | 7.051 | 2.166% | 2s | 10.762 | 4.102% | 8s | | CaDA‡ | 11.760 | 0.771% | 2s | 19.435 | 1.439% | 9s |
| | CaDA | 7.062 | 2.339% | 1s | 10.741 | 3.900% | 8s | | CaDA | 11.777 | 0.914% | 2s | 19.497 | 1.762% | 9s |
| | CCL | **7.009** | **1.569%** | 5s | **10.681** | **3.323%** | 20s | | CCL | **11.721** | **0.442%** | 6s | **19.346** | **0.977%** | 22s |
| OVRPL | HGS-PyVRP | 6.507 | * | 10.4m | 9.724 | * | 20.8m | OVRPLTW | HGS-PyVRP | 10.510 | * | 10.4m | 16.926 | * | 20.8m |
| | CaDA‡ | 6.671 | 2.475% | 2s | 10.122 | 4.052% | 8s | | CaDA‡ | 10.613 | 0.961% | 2s | 17.226 | 1.752% | 9s |
| | CaDA | 6.680 | 2.623% | 1s | 10.093 | 3.773% | 8s | | CaDA | 10.631 | 1.133% | 1s | 17.280 | 2.073% | 9s |
| | CCL | **6.637** | **1.968%** | 5s | **10.067** | **3.495%** | 20s | | CCL | **10.569** | **0.546%** | 6s | **17.123** | **1.143%** | 22s |
| VRPB | HGS-PyVRP | 9.687 | * | 10.4m | 14.377 | * | 20.8m | VRPBTW | HGS-PyVRP | 18.292 | * | 10.4m | 29.467 | * | 20.8m |
| | CaDA‡ | 9.960 | 2.800% | 2s | 14.960 | 4.038% | 8s | | CaDA‡ | 18.500 | 1.117% | 2s | 30.059 | 1.999% | 9s |
| | CaDA | 9.979 | 3.010% | 1s | 14.910 | 3.721% | 8s | | CaDA | 18.543 | 1.361% | 1s | 30.174 | 2.390% | 9s |
| | CCL | **9.916** | **2.352%** | 5s | **14.882** | **3.526%** | 19s | | CCL | **18.430** | **0.738%** | 6s | **29.911** | **1.494%** | 21s |
| VRPBL | HGS-PyVRP | 10.186 | * | 10.4m | 14.779 | * | 20.8m | VRPBLTW | HGS-PyVRP | 18.361 | * | 10.4m | 29.026 | * | 20.8m |
| | CaDA‡ | 10.543 | 3.461% | 2s | 15.525 | 5.001% | 8s | | CaDA‡ | 18.848 | 1.376% | 2s | 30.520 | 2.359% | 9s |
| | CaDA | 10.576 | 3.776% | 1s | 15.490 | 4.771% | 8s | | CaDA | 18.894 | 1.623% | 1s | 30.620 | 2.700% | 9s |
| | CCL | **10.484** | **2.883%** | 5s | **15.407** | **4.219%** | 19s | | CCL | **18.773** | **0.976%** | 6s | **30.366** | **1.842%** | 21s |
| VRPL | HGS-PyVRP | 10.587 | * | 10.4m | 15.766 | * | 20.8m | VRPLTW | HGS-PyVRP | 16.356 | * | 10.4m | 25.757 | * | 20.8m |
| | CaDA‡ | 10.731 | 1.333% | 2s | 16.057 | 1.847% | 8s | | CaDA‡ | 16.669 | 1.879% | 2s | 26.540 | 2.995% | 9s |
| | CaDA | 10.749 | 1.505% | 1s | 16.036 | 1.725% | 8s | | CaDA | 16.709 | 2.130% | 1s | 26.631 | 3.358% | 9s |
| | CCL | **10.710** | **1.145%** | 5s | **16.009** | **1.561%** | 19s | | CCL | **16.579** | **1.333%** | 6s | **26.366** | **2.321%** | 20s |
| Avg. | HGS-PyVRP | 8.455 | * | 10.4m | 12.584 | * | 20.8m | Avg. | HGS-PyVRP | 14.175 | * | 10.4m | 22.730 | * | 20.8m |
| | CaDA‡ | 8.646 | 2.256% | 2s | 13.022 | 3.595% | 8s | | CaDA‡ | 14.380 | 1.172% | 2s | 23.314 | 2.033% | 9s |
| | CaDA | 8.662 | 2.437% | 1s | 12.993 | 3.372% | 8s | | CaDA | 14.409 | 1.366% | 1s | 23.394 | 2.381% | 9s |
| | CCL | **8.609** | **1.802%** | 5s | **12.950** | **3.013%** | 19s | | CCL | **14.319** | **0.749%** | 6s | **23.187** | **1.476%** | 21s |

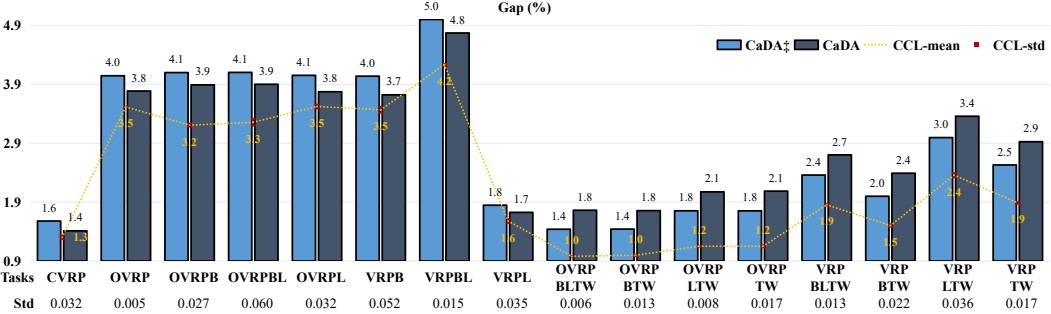

Figure 4: Error-bar analysis of CCL under N=100.

## C  ADDITIONAL ANALYSES AND DISCUSSIONS

### C.1  COMPARISON WITH NEURAL SOTA METHODS

#### C.1.1  COMPARISON BETWEEN RE-IMPLEMENTED SOTA AND REPORTED SOTA

**Main Results.** We compare CCL against the reported SOTA method (CaDA‡ (Li et al., 2025)) and our re-implemented one (CaDA (Li et al., 2025)), with the strong heuristic baseline HGS-PyVRP (Wouda et al., 2024) included for reference. Table 5 presents the results for all 16 in-distribution tasks, and the overall average scores across these tasks both with and without the TW constraint. Under the N=100

Table 6: Improvement of CCL$^\dagger$ over CaDA$^\dagger$.

| Tasks | $\Delta$ | P-value | Tasks | $\Delta$ | P-value | Tasks | $\Delta$ | P-value | Tasks | $\Delta$ | P-value |
|---|---|---|---|---|---|---|---|---|---|---|---|
| CVRP | 8.16% | 2.7e-05 | OVRP | 29.21% | 1.2e-76 | VRPBLTW | 41.61% | 2.2e-68 | OVRPBLTW | 51.19% | 6.2e-68 |
| VRPB | 20.14% | 5.0e-42 | OVRPB | 29.86% | 1.4e-62 | VRPBTW | 46.84% | 4.0e-81 | OVRPBTW | 50.63% | 2.8e-68 |
| VRPBL | 20.03% | 2.0e-39 | OVRPBL | 31.05% | 4.7e-68 | VRPLTW | 41.39% | 2.2e-97 | OVRPLTW | 51.50% | 2.9e-79 |
| VRPL | 7.66% | 2.4e-04 | OVRPL | 28.69% | 3.5e-74 | VRPTW | 45.72% | 2.9e-99 | OVRPTW | 50.85% | 3.2e-75 |

Table 7: Per-task results on 32 unseen out-of-distribution tasks ($N$=50).

| Tasks | RF-TE | | | CaDA | | | CaDA$^\dagger$ | | | CCL | | | CCL$^\dagger$ | | |
|---|---|---|---|---|---|---|---|---|---|---|---|---|---|---|---|
| | Obj. $\downarrow$ | Gap $\downarrow$ | Time $\downarrow$ | Obj. $\downarrow$ | Gap $\downarrow$ | Time $\downarrow$ | Obj. $\downarrow$ | Gap $\downarrow$ | Time $\downarrow$ | Obj. $\downarrow$ | Gap $\downarrow$ | Time $\downarrow$ | Obj. $\downarrow$ | Gap $\downarrow$ | Time $\downarrow$ |
| VRPMB | 9.879 | 8.861% | 1s | 9.781 | 7.749% | 2s | **9.722** | **7.097%** | **3s** | 9.943 | 9.538% | 3s | 9.940 | 9.486% | 4s |
| MDCVRP | 12.559 | 56.957% | 2s | 12.335 | 54.083% | 2s | 13.593 | 70.078% | 4s | 10.829 | 34.846% | 4s | **10.554** | **31.361%** | 6s |
| MDOVRP | 6.876 | 29.051% | 1s | 6.825 | 28.069% | 2s | **6.260** | **17.329%** | **3s** | 6.413 | 20.155% | 4s | 6.286 | 17.719% | 5s |
| MDVRPB | 12.725 | 60.956% | 2s | 12.654 | 60.054% | 2s | 12.190 | 54.066% | 3s | 11.679 | 47.450% | 4s | **11.001** | **38.750%** | 5s |
| MDVRPL | 12.618 | 57.370% | 2s | 12.426 | 54.988% | 2s | 13.433 | 67.697% | 4s | 11.278 | 40.248% | 4s | **10.912** | **35.667%** | 6s |
| OVRPMB | 6.949 | 13.690% | 1s | 6.872 | 12.440% | 1s | **6.819** | **11.570%** | **3s** | 6.917 | 13.149% | 3s | 6.854 | 12.124% | 4s |
| VRPMBL | 10.239 | 8.050% | 1s | 10.163 | 7.215% | 2s | **10.061** | **6.130%** | **3s** | 10.376 | 9.456% | 3s | 10.229 | 7.898% | 5s |
| MDOVRPB | 7.556 | 31.861% | 1s | 7.622 | 33.009% | 2s | 6.999 | 22.032% | 3s | 6.876 | 19.787% | 4s | **6.738** | **17.332%** | 5s |
| MDOVRPL | 6.871 | 28.946% | 1s | 6.807 | 27.746% | 2s | **6.270** | **17.504%** | **3s** | 6.442 | 20.716% | 4s | 6.303 | 18.070% | 5s |
| MDVRPBL | 12.831 | 61.175% | 2s | 12.585 | 58.011% | 2s | 12.017 | 50.770% | 3s | 11.483 | 43.846% | 4s | **11.208** | **40.413%** | 6s |
| MDVRPMB | 12.856 | 76.544% | 2s | 12.493 | 71.550% | 2s | 12.046 | 65.185% | 3s | 11.540 | 58.118% | 4s | **11.032** | **51.055%** | 6s |
| MDVRPTW | 17.818 | 48.941% | 2s | 16.971 | 41.739% | 2s | 17.581 | 46.865% | 4s | 16.107 | 34.403% | 4s | **15.475** | **29.004%** | 6s |
| OVRPMBL | 6.949 | 13.686% | 1s | 6.871 | 12.423% | 1s | 6.818 | 11.563% | 3s | 6.932 | 13.404% | 3s | **6.819** | **11.555%** | 4s |
| VRPMBTW | 17.298 | 8.074% | 1s | 17.198 | 7.434% | 2s | 17.282 | 7.954% | 3s | **16.988** | **6.099%** | 3s | 17.158 | 7.142% | 5s |
| MDOVRPBL | 7.550 | 31.772% | 1s | 7.606 | 32.713% | 2s | 6.996 | 21.966% | 3s | 6.858 | 19.463% | 4s | **6.827** | **18.939%** | 6s |
| MDOVRPMB | 7.617 | 47.411% | 1s | 7.541 | 45.953% | 2s | 6.845 | 32.344% | 3s | 6.720 | 29.826% | 4s | **6.519** | **25.920%** | 5s |
| MDOVRPTW | 10.618 | 34.976% | 1s | 10.204 | 29.610% | 2s | 10.407 | 32.287% | 4s | 9.783 | 24.150% | 4s | **9.632** | **22.146%** | 5s |
| MDVRPBTW | 18.591 | 37.364% | 2s | 19.025 | 40.629% | 2s | 19.977 | 47.721% | 4s | 18.425 | 36.029% | 4s | **17.645** | **30.121%** | 5s |
| MDVRPLTW | 18.127 | 51.276% | 2s | 17.125 | 42.765% | 2s | 17.965 | 49.851% | 4s | 16.769 | 39.728% | 5s | **16.126** | **34.232%** | 6s |
| MDVRPMBL | 12.744 | 74.112% | 2s | 12.434 | 69.825% | 2s | 11.647 | 58.969% | 3s | 11.474 | 56.409% | 4s | **10.993** | **49.737%** | 6s |
| OVRPMBTW | 11.132 | 6.265% | 1s | 11.087 | 5.849% | 2s | 11.121 | 6.160% | 3s | 10.966 | 4.658% | 3s | **10.855** | **3.617%** | 5s |
| VRPMBLTW | 17.597 | 7.982% | 1s | 17.495 | 7.337% | 2s | 17.559 | 7.728% | 3s | 17.514 | 7.433% | 3s | **17.402** | **6.710%** | 5s |
| MDOVRPBTW | 11.399 | 32.332% | 2s | 11.190 | 29.807% | 2s | 11.423 | 32.600% | 4s | 10.774 | 24.830% | 4s | **10.313** | **19.359%** | 5s |
| MDOVRPLTW | 10.599 | 34.731% | 2s | 10.196 | 29.501% | 2s | 10.408 | 32.286% | 4s | 9.721 | 23.331% | 4s | **9.704** | **23.040%** | 5s |
| MDOVRPMBL | 7.602 | 47.108% | 1s | 7.540 | 45.940% | 2s | 6.848 | 32.401% | 3s | 6.733 | 30.088% | 4s | **6.548** | **26.474%** | 5s |
| MDVRPBLTW | 19.048 | 40.583% | 2s | 19.243 | 42.015% | 2s | 20.076 | 48.201% | 4s | 18.544 | 36.702% | 5s | **17.612** | **29.641%** | 7s |
| MDVRPMBTW | 17.830 | 48.485% | 2s | 18.394 | 53.345% | 2s | 19.327 | 61.083% | 4s | 17.103 | 42.314% | 5s | **16.729** | **39.084%** | 5s |
| OVRPMBLTW | 11.138 | 6.317% | 1s | 11.090 | 5.875% | 2s | 11.116 | 6.109% | 3s | 10.993 | 4.922% | 3s | **10.835** | **3.427%** | 5s |
| MDOVRPBLTW | 11.389 | 32.207% | 2s | 11.185 | 29.743% | 2s | 11.391 | 32.216% | 4s | 10.767 | 24.749% | 4s | **10.350** | **19.783%** | 5s |
| MDOVRPMBTW | 11.055 | 40.153% | 2s | 10.833 | 37.232% | 2s | 11.150 | 41.336% | 4s | 10.171 | 28.688% | 4s | **10.110** | **27.872%** | 5s |
| MDVRPMBLTW | 18.222 | 51.554% | 2s | 18.511 | 54.026% | 2s | 19.362 | 61.119% | 4s | 17.940 | 49.174% | 5s | **16.825** | **39.635%** | 6s |
| MDOVRPMBLTW | 11.054 | 40.136% | 2s | 10.818 | 37.044% | 2s | 11.135 | 41.125% | 4s | 10.203 | 29.114% | 4s | **9.804** | **23.943%** | 6s |
| Avg. Gap | | 36.529% | | | 34.866% | | | 34.417% | | | 27.588% | | | **24.102%** | |
| # Best (Best/Total) | | 0/32 | | | 0/32 | | | 5/32 | | | 1/32 | | | 26/32 | |

without TW, CaDA outperforms the reported CaDA$^\ddagger$, while in all other settings it is slightly inferior to CaDA$^\ddagger$. In contrast, CCL consistently surpasses both CaDA and CaDA$^\ddagger$ across all 16 tasks.

**Error Bar Analysis.** Since the testing update probability $P_{ts}$ is set to 0.5 for $N$=100, we further analyze the error bars of CCL under different random seeds. Fig. 4 plots the mean gap and its standard deviation over three independent test runs of CCL. Across all 16 tasks, the standard deviation of CCL's gap remains tightly bound between 0.005% and 0.060%. Visually, the error bars in Figure 5 are negligible compared to the performance difference between CCL and CaDA/CaDA$^\ddagger$, indicating that the choice of seed has minimal impact on test-time results.

### C.1.2 STATISTICAL SIGNIFICANCE

We continue to include t-tests to assess statistical significance. We first collected the gap values of 1,000 test instances from both CCL$^\dagger$ and the strongest SOTA CaDA$^\dagger$ from Table 1, then we report the improvement percentages ($\Delta$) along with the corresponding p-values (shown in Table 6). Here, the improvements are computed as the average gap reductions of CCL$^\dagger$ over CaDA$^\dagger$, *i.e.*, $-(\text{Gap}(\text{CCL}^\dagger) - \text{Gap}(\text{CaDA}^\dagger))/\text{Gap}(\text{CaDA}^\dagger) \times 100\%$. Across all 16 tasks, CCL achieves 7-51% improvement. In particular, OVRPBLTW, OVRPBTW, OVRPLTW, and OVRPTW exceed 50%. All p-values are below 0.001, indicating that these gains are statistically significant.

### C.1.3 DETAILED RESULTS ON UNSEEN OUT-OF-DISTRIBUTION TASKS

We provide per-task results on 32 unseen out-of-distribution tasks. Each method is evaluated in a zero-shot setting (*i.e.*, directly tested without fine-tuning). For the $N$=50 setting, the test-time update probability is set to $P_{ts}$=0.15, and for $N$=100, it is set to $P_{ts}$=0.02. Table 7 and Table 8 present the full results under the $N$=50 and $N$=100 settings, respectively. For each task, we report the objective (Obj.), performance gap (Gap), and inference time (Time). Additionally, the bottom rows summarize

Table 8: Per-task results on 32 unseen out-of-distribution tasks ($N$=100).

| Tasks | RF-TE | | | CaDA | | | CaDA† | | | CCL | | | CCL† | | |
|---|---|---|---|---|---|---|---|---|---|---|---|---|---|---|---|
| | Obj. ↓ | Gap ↓ | Time ↓ | Obj. ↓ | Gap ↓ | Time ↓ | Obj. ↓ | Gap ↓ | Time ↓ | Obj. ↓ | Gap ↓ | Time ↓ | Obj. ↓ | Gap ↓ | Time ↓ |
| VRPMB | 14.888 | 10.189% | 8s | 14.710 | 8.822% | 8s | **14.652** | **8.399%** | 13s | 15.322 | 13.438% | 14 | 14.936 | 10.530% | 18s |
| MDCVRP | 19.684 | 67.107% | 10s | 20.628 | 75.392% | 11s | 20.964 | 78.117% | 18s | **16.769** | **41.675%** | 17s | 16.834 | 42.308% | 24s |
| MDOVRP | 10.683 | 34.368% | 9s | 10.605 | 33.387% | 10s | 9.849 | 23.695% | 15s | 10.095 | 26.715% | 15s | **9.753** | **22.409%** | 20s |
| MDVRPB | 18.721 | 61.761% | 9s | 19.494 | 68.604% | 11s | 18.780 | 62.221% | 17s | **18.185** | **56.977%** | 19s | 18.625 | 60.931% | 25s |
| MDVRPL | 20.100 | 70.498% | 10s | 20.867 | 77.336% | 12s | 21.001 | 78.277% | 18s | **17.494** | **47.699%** | 24s | 18.407 | 55.828% | 29s |
| OVRPMB | 10.711 | 18.899% | 7s | 10.490 | 16.449% | 8s | **10.468** | **16.205%** | 13s | 10.780 | 19.642% | 14s | 10.744 | 19.235% | 19s |
| VRPMBL | 15.198 | 10.375% | 7s | 15.016 | 9.011% | 8s | **14.949** | **8.536%** | 13s | 15.761 | 14.464% | 14s | 15.538 | 12.835% | 18s |
| MDOVRPB | 11.752 | 35.659% | 9s | 11.790 | 36.137% | 10s | 11.104 | 28.078% | 16s | 10.959 | 26.274% | 16s | **10.533** | **21.340%** | 21s |
| MDOVRPL | 10.703 | 34.620% | 9s | 10.574 | 32.985% | 10s | 9.854 | 23.749% | 15s | 10.272 | 29.033% | 16s | **9.718** | **21.967%** | 20s |
| MDVRPBL | 19.606 | 68.898% | 11s | 19.827 | 70.875% | 13s | 18.693 | 60.860% | 18s | 18.446 | 58.520% | 21s | **17.588** | **51.069%** | 28s |
| MDVRPMB | 18.698 | 76.325% | 9s | 19.458 | 83.817% | 11s | 18.374 | 73.239% | 18s | 18.997 | 79.149% | 18s | **17.544** | **65.282%** | 23s |
| MDVRPTW | 29.468 | 53.283% | 11s | **25.955** | **34.783%** | 10s | 28.751 | 49.517% | 17s | 26.901 | 39.557% | 18s | 29.016 | 50.829% | 28s |
| OVRPMBL | 10.709 | 18.877% | 7s | 10.486 | 16.399% | 8s | **10.470** | **16.225%** | 13s | 10.883 | 20.748% | 14s | 10.749 | 19.284% | 19s |
| VRPMBTW | 28.256 | 10.840% | 8s | 28.317 | 11.074% | 9s | 28.310 | 11.038% | 14s | 28.100 | 10.239% | 15s | **27.971** | **9.722%** | 20s |
| MDOVRPBL | 11.761 | 35.771% | 9s | 11.788 | 36.106% | 10s | 11.105 | 28.077% | 16s | 10.983 | 26.545% | 16s | **10.511** | **21.090%** | 21s |
| MDOVRPMB | 11.748 | 53.650% | 9s | 11.692 | 53.010% | 10s | 10.945 | 43.077% | 16s | 10.876 | 42.044% | 17s | **10.431** | **36.189%** | 21s |
| MDOVRPTW | 18.299 | 41.356% | 10s | 17.205 | 32.804% | 10s | 17.636 | 36.201% | 17s | 17.009 | 31.200% | 17s | **16.733** | **29.076%** | 23s |
| MDVRPBTW | 30.681 | 39.926% | 10s | **30.176** | **37.559%** | 10s | 31.716 | 44.719% | 19s | 30.677 | 39.759% | 19s | 31.269 | 42.473% | 30s |
| MDVRPLTW | 29.640 | 53.976% | 11s | **26.200** | **35.862%** | 10s | 29.327 | 52.284% | 17s | 27.977 | 45.055% | 23s | 29.593 | 53.413% | 31s |
| MDVRPMBL | 19.216 | 80.892% | 11s | 19.460 | 83.444% | 13s | 18.183 | 71.054% | 18s | 18.479 | 73.647% | 19s | **18.112** | **70.333%** | 32s |
| OVRPMBTW | 18.449 | 8.724% | 8s | 18.478 | 8.901% | 9s | 18.430 | 8.607% | 15s | 18.427 | 8.590% | 16s | **18.211** | **7.321%** | 21s |
| VRPMBTW | 28.604 | 10.805% | 8s | 28.658 | 11.002% | 9s | 28.641 | 10.925% | 14s | **28.374** | **9.907%** | 15s | 28.582 | 10.698% | 20s |
| MDOVRPBTW | 19.590 | 36.897% | 10s | 19.305 | 34.918% | 10s | 19.341 | 35.157% | 18s | 18.764 | 30.960% | 17s | **18.265** | **27.473%** | 26s |
| MDOVRPLTW | 18.232 | 40.817% | 10s | 17.221 | 32.923% | 10s | 17.665 | 36.428% | 17s | 16.838 | 29.857% | 18s | **16.707** | **28.872%** | 23s |
| MDOVRPMBL | 11.751 | 53.691% | 10s | 11.670 | 52.726% | 10s | 10.931 | 42.890% | 16s | 10.830 | 41.416% | 16s | **10.415** | **35.973%** | 21s |
| MDVRPBLTW | 31.044 | 41.408% | 10s | **30.537** | **39.033%** | 10s | 32.239 | 46.923% | 18s | 30.667 | 39.535% | 24s | 34.176 | 55.800% | 32s |
| MDVRPMBTW | 29.650 | 54.395% | 10s | 29.383 | 53.001% | 10s | 30.722 | 60.046% | 18s | **28.388** | **47.661%** | 18s | 30.844 | 60.585% | 26s |
| OVRPMBLTW | 18.452 | 8.739% | 8s | 18.476 | 8.887% | 9s | 18.415 | 8.516% | 15s | **18.404** | **8.443%** | 16s | 18.476 | 8.877% | 20s |
| MDOVRPBLTW | 19.553 | 36.632% | 10s | 19.341 | 35.154% | 10s | 19.349 | 35.212% | 18s | 18.853 | 31.599% | 17s | **18.194** | **26.963%** | 25s |
| MDOVRPMBTW | 18.888 | 46.216% | 10s | 18.735 | 45.019% | 10s | 18.761 | 45.232% | 17s | 17.823 | 37.814% | 18s | **17.638** | **36.412%** | 23s |
| MDVRPMBLTW | 29.825 | 55.132% | 10s | 29.611 | 53.989% | 11s | 31.157 | 62.147% | 18s | **29.000** | **50.703%** | 23s | 31.057 | 61.451% | 35s |
| MDOVRPMBLTW | 18.846 | 45.873% | 10s | 18.770 | 45.286% | 10s | 18.787 | 45.427% | 17s | 17.803 | 37.665% | 17s | **17.667** | **36.646%** | 24s |
| Avg. Gap | | 41.144% | | | 39.834% | | | 39.096% | | | 34.892% | | | **34.788%** | |
| # Best (Best/Total) | | 0/32 | | | 4/32 | | | 4/32 | | | 7/32 | | | **17/32** | |

each method's average gap and the number of tasks where it achieves the best performance, reported in the format "# Best (best/total)". Across both the $N$=50 and $N$=100 settings, CCL† consistently achieves the lowest average gap, demonstrating strong generalization to unseen out-of-distribution tasks. In terms of per-task performance, CCL and CCL† together outperform all baselines on 27 out of 32 tasks for $N$=50, and on 24 out of 32 tasks for $N$=100, further highlighting the robustness and effectiveness of our method across different problem scales.

## C.2 COMPARISON WITH TRADITIONAL SOLVER

We follow RouteFinder (Berto et al., 2024; 2025) and CaDA (Li et al., 2025) in using HGS-PyVRP (Wouda et al., 2024) as a strong traditional solver. Moreover, we compare our method against additional traditional solvers, including Gurobi (Gurobi Optimization, 2024) and LKH (Lin and Kernighan, 1973) on CVRP instances with 50 customers. The total times, denoted as "Time", are accumulated over 1000 instances, which are exactly the same as the ones used in (Berto et al., 2024; 2025; Li et al., 2025). We also provide the average per-instance time for reference, denoted as "Avg. Time". As shown in Table 9, the results of HGS-PyVRP are taken from (Li et al., 2025), while the results of Gurobi and LKH are obtained using a 32-core CPU. Based on this, Gurobi further uses 4 threads per CPU core, enabling 4×32 instances to be solved in parallel. LKH is executed for 10 runs, with a 10-second time limit per instance. We set a 15-minute limit on Gurobi and report its generated (approximate) solutions. These results show that CCL achieves performance comparable to traditional solvers, while its total inference time is approximately 10×, 1,200×, and 100× faster than LKH, Gurobi, and HGS-PyVRP, respectively (corresponding per-instance speedups of 5×, 75×, and 50×). This demonstrates that learning-based models are practical for real-time applications, especially when solving multiple VRP instances simultaneously. Moreover, in multi-task scenarios, CCL can learn generalizable patterns across different VRP variants, without requiring experts to manually design heuristics. Once trained, the model can solve 48 VRP variants without re-training, which will broaden its practical deployment. These findings show that both traditional algorithms and our learning-based method have their own merits and demerits. Research in either direction provides important insights for the VRP community.

Table 9: Comparison with traditional solvers.

| Methods | Obj. ↓ | Time ↓ | Avg. Time ↓ |
|---|---|---|---|
| HGS-PyVRP | 10.372 | 10m | 10.0s |
| Gurobi-15m | 10.568 | 120m | 15.0m |
| LKH | 10.392 | 63s | 1.1s |
| CCL† | 10.463 | 6s | 0.2s |

## C.3 ABLATION RESULTS

### C.3.1 ABLATION RESULTS WITHOUT RELD

We conducted ablation studies to examine whether the effectiveness of our method depends on ReLD (Huang et al., 2025). Starting from CCL$^\dagger$, we remove ReLD, RGCR, TSNR, and both RGCR and TSNR. The corresponding average results across 16 in-distribution tasks are presented in Table 10. The results show that both RGCR and TSNR remain effective even without ReLD. Moreover, CCL reduces the average gap

Table 10: Ablation on CCL (variant w/o ReLD).

| Methods | Obj.↓ | Gap↓ | Time↓ |
|---|---|---|---|
| CCL$^\dagger$ | 11.447 | 1.10% | 6.5s |
| - ReLD (CCL) | 11.464 | 1.28% | 5.4s |
| - RGCR | 11.477 | 1.40% | 4.6s |
| - TSNR | 11.518 | 1.76% | 2.8s |
| - TSNR - RGCR | 11.529 | 1.88% | 2.0s |

by 0.6% compared with CCL-TSNR-RGCR, whereas CCL$^\dagger$ provides only a 0.18% improvement over CCL. This indicates that the main performance improvement comes from CCL rather than ReLD.

### C.3.2 ABLATION DETAILS WITHIN RGCR

To validate the effectiveness of RGCR, we test 16 in-domain VRP variants, each with 1,000 instances, and compare RGCR with four alternatives: (1) "Concat Attributes", which directly concatenates the constraint attributes; (2) "Concat Embeddings", which embeds each constraint into a high-dimensional space and concatenates them; (3) "+ Random Scores" employ random importance weights as the correlation scores; and (4) "+ Cosine Similarity" uses cosine similarity to measure the correlation scores. We present both the model complexity and performance results on the left of Fig. 2. Specifically, "# Params" denotes the total number of parameters in the encoder and decoder, and "Time" is the accumulated inference time over 1,000 instances. "Avg. Gap" denotes the average gap across all 16 tasks, while "w/ TW" and "w/o TW" refer to the subsets of 8 tasks with and without time-window constraints, respectively. Compared with concatenating attributes, RGCR achieves strong performance while increasing the model size by only 0.1M parameters and adding 1.3s to inference time. Compared to random weights, RGCR shows modest performance in average gap across the 16 tasks, but demonstrates clear superiority on tasks with time windows. These results demonstrate that RGCR benefits more on complex tasks than on simpler ones. This may be attributed to the fact that tasks without time windows often include a lot of padding information, which may introduce some noise during model training. Moreover, RGCR introduces no additional parameters compared to the "Concat Embeddings" setting, yet still reduces the average gap by 0.041%. This indicates that the gains arise from improved constraint prioritization rather than model capacity.

### C.3.3 DESIGN CHOICE OF ATTENTION STRATEGY IN TSNR

In TSNR, we adopt a cross-attention mechanism, where the node embedding serves as the query and the unified node-constraint embedding as the key and value. The following theoretical analysis and empirical results show that this approach reduces computational complexity compared to the vanilla Transformer, which applies self-attention on the unified embedding. Specifically, the unified embedding has dimension $(N+(N+1))\times D$, while the context and node embeddings have dimensions $N \times D$ and $(N + 1) \times D$, respectively. Consequently, self-attention computes $(2N + 1) \times (2N + 1)$ attention weights, whereas cross-attention computes only $(N + 1) \times (2N + 1)$. As shown in Table 11, cross-attention achieves comparable performance while reducing inference time. For example, across tasks with TW (*i.e.*, the right half of Table 11), it narrows the gap from 0.727% to 0.689% and reduces inference time by 1s.

## C.4 PERFORMANCE COMPARISON UNDER MATCHED INFERENCE TIME

To further validate that the effectiveness of CCL is not merely due to increased inference time, we introduce a heavy decoder variant of CaDA, denoted as CaDA-HD. This version deepens the original 1-layer Transformer decoder to 4 layers, resulting in an inference time that is comparable to CCL. Table 12 presents the performance comparison between CaDA-HD and CCL under similar inference budgets. These results indicate that CCL and CaDA-HD perform similarly on tasks without time windows (TW), with CaDA-HD sometimes showing slightly better results. In contrast, on tasks with TW, CCL consistently outperforms CaDA-HD by a significant margin. This suggests that CCL's

Table 11: Performance comparison under different attention mechanisms.

| Tasks | Cross-Attention | | | Self-Attention | | | Tasks | Cross-Attention | | | Self-Attention | | |
|---|---|---|---|---|---|---|---|---|---|---|---|---|---|
| | Obj. ↓ | Gap ↓ | Time ↓ | Obj. ↓ | Gap ↓ | Time ↓ | | Obj. ↓ | Gap ↓ | Time ↓ | Obj. ↓ | Gap ↓ | Time ↓ |
| CVRP | 10.463 | 0.881% | **6s** | **10.461** | **0.867%** | 7s | VRPTW | **16.177** | **0.907%** | 7s | 16.186 | 0.955% | 8s |
| OVRP | **6.610** | **1.566%** | **6s** | 6.611 | 1.582% | 7s | OVRPTW | **10.564** | **0.506%** | 7s | 10.568 | 0.540% | 8s |
| VRPB | 9.875 | 1.921% | **6s** | **9.873** | **1.896%** | 7s | VRPBTW | **18.419** | **0.678%** | 7s | 18.427 | 0.719% | 8s |
| VRPL | 10.698 | 1.027% | **6s** | **10.694** | **0.993%** | 7s | VRPLTW | **16.556** | **1.192%** | 7s | 16.564 | 1.246% | 8s |
| OVRPB | 6.992 | 1.344% | **6s** | **6.992** | **1.337%** | 7s | OVRPBTW | **11.718** | **0.416%** | 7s | 11.721 | 0.444% | 8s |
| OVRPL | **6.610** | **1.569%** | **6s** | 6.611 | 1.582% | 7s | OVRPLTW | **10.564** | **0.501%** | 7s | 10.565 | 0.517% | 8s |
| VRPBL | **10.440** | **2.450%** | **6s** | 10.445 | 2.489% | 7s | VRPBLTW | **18.758** | **0.899%** | 7s | 18.769 | 0.952% | 8s |
| OVRPBL | 6.992 | 1.335% | **6s** | **6.992** | **1.330%** | 7s | OVRPBLTW | **11.718** | **0.414%** | 7s | 11.721 | 0.446% | 8s |
| Avg. | 8.585 | 1.512% | **6s** | 8.585 | **1.510%** | 7s | Avg. | **14.309** | **0.689%** | 7s | 14.315 | 0.727% | 8s |

Table 12: Performance comparison under matched inference time.

| Tasks | CaDA-HD | | | CCL | | | Tasks | CaDA-HD | | | CCL | | |
|---|---|---|---|---|---|---|---|---|---|---|---|---|---|
| | Obj. ↓ | Gap ↓ | Time ↓ | Obj. ↓ | Gap ↓ | Time ↓ | | Obj. ↓ | Gap ↓ | Time ↓ | Obj. ↓ | Gap ↓ | Time ↓ |
| CVRP | **10.468** | **0.926%** | 6s | 10.473 | 0.977% | 5s | VRPTW | 16.293 | 1.631% | 5s | **16.190** | **0.979%** | 5s |
| OVRP | **6.635** | **1.937%** | 5s | 6.636 | 1.957% | 5s | OVRPTW | 10.615 | 0.986% | 5s | **10.569** | **0.543%** | 6s |
| VRPB | **9.908** | **2.267%** | 5s | 9.916 | 2.352% | 5s | VRPBTW | 18.529 | 1.280% | 5s | **18.430** | **0.738%** | 6s |
| VRPL | **10.704** | **1.091%** | 5s | 10.710 | 1.145% | 5s | VRPLTW | 16.676 | 1.930% | 5s | **16.579** | **1.333%** | 6s |
| OVRPB | 7.019 | 1.725% | 5s | **7.008** | **1.568%** | 5s | OVRPBTW | 11.772 | 0.874% | 6s | **11.721** | **0.436%** | 6s |
| OVRPL | **6.633** | **1.908%** | 5s | 6.637 | 1.968% | 5s | OVRPLTW | 10.615 | 0.986% | 5s | **10.569** | **0.546%** | 6s |
| VRPBL | **10.480** | **2.842%** | 5s | 10.484 | 2.883% | 5s | VRPBLTW | 18.872 | 1.498% | 5s | **18.773** | **0.976%** | 6s |
| OVRPBL | 7.020 | 1.721% | 5s | **7.009** | **1.569%** | 5s | OVRPBLTW | 11.771 | 0.864% | 6s | **11.721** | **0.442%** | 6s |
| Avg. | **8.608** | **1.802%** | 5s | 8.609 | 1.802% | 5s | Avg. | 14.393 | 1.256% | 5s | **14.319** | **0.749%** | 6s |

design is particularly effective in handling temporally constrained problems, and its advantage is not merely a result of longer inference time.

## C.5 CONVERGENCE ANALYSIS

Fig. 5 shows the training loss of CCL and CaDA. CCL achieves faster convergence in the early epochs and reaches a lower final loss compared to CaDA. For instance, at epoch 50, the loss of CCL is 0.0129, while CaDA is 0.0198. By the end of training, CCL attains 0.0090 versus 0.0129 for CaDA. These results indicate that CCL contributes to more efficient and effective training, yielding both faster convergence and improved final performance.

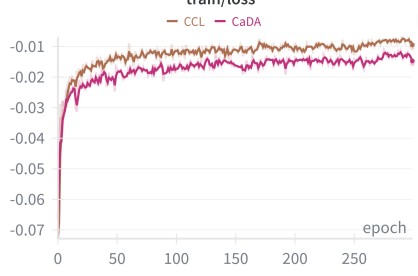

Figure 5: Training loss convergence.

## C.6 GENERALIZATION OF CCL ON ADDITIONAL SCENARIOS

Same as RouteFinder (Berto et al., 2024; 2025), CaDA (Li et al., 2025), MTPOMO (Liu et al., 2024), and MvMoE (Zhou et al., 2024a), our work also focuses on solving routing problems solely. Moreover, the design of CCL can be useful for other decision-making problems. We conduct a preliminary experiment on the Flexible Flow Shop Problem (FFSP). When assigning an operation to a machine, we incorporated TSNR to allow operation embeddings to integrate information from the current machine, thereby updating the operation's state. Results in Fig. 6 (a) show that the method converges, but training can become slightly unstable from the middle to late stages. This suggests that certain modifications to TSNR may be needed for the best performance, for example, to filter out irrelevant information in the machine embeddings that does not contribute to subsequent decision-making.

We also conduct an experiment using graph-structured inputs instead of coordinates. Specifically, each node's coordinates were replaced with a vector of distances to all other nodes, which is then concatenated with demand and other attributes to form the node inputs. We retrain CCL across 16 tasks, each with 50 customers. During the 300 training epochs, we report the training loss and the validation average objective length across 128 CVRP instances (also with 50 customers). Fig. 6 (b) shows that both the training loss and the validation scores of CCL converge quickly within the first 50 epochs, suggesting the potential of CCL for graph-structured VRPs. To further investigate the effectiveness of RGCR and TSNR, we apply this setting to retrain the corresponding baseline model

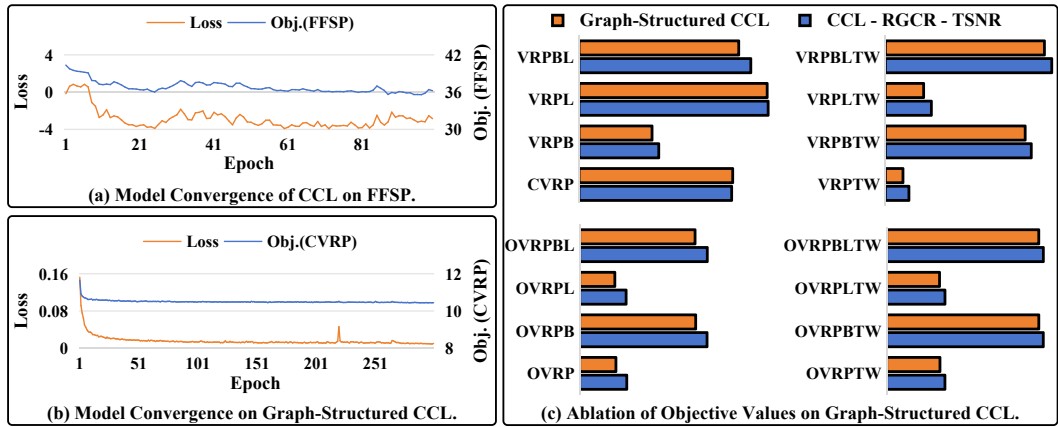

Figure 6: Generalizing CCL on the flexible flow shop problem (FFSP) and graph-structured input.

Table 13: Per-instance results on small-scale real-world instances. In CVRP, "X-n***" denotes the customer number of each instance, while each VRPTW instance has 100 customers.

| CVRP | | RF-TE | | CaDA | | CCL | | CCL-Ens | | VRPTW | | RF-TE | | CaDA | | CCL | |
|---|---|---|---|---|---|---|---|---|---|---|---|---|---|---|---|---|---|
| Instances | Opt. | Obj. ↓ | Gap ↓ | Obj. ↓ | Gap ↓ | Obj. ↓ | Gap ↓ | Obj. ↓ | Gap ↓ | Instances | Opt. | Obj. ↓ | Gap ↓ | Obj. ↓ | Gap ↓ | Obj. ↓ | Gap ↓ |
| X-n101-k25 | 27591 | 29087 | 5.422% | 28765 | 4.255% | 28765 | 4.255% | **28727** | **4.117%** | R101 | 1638 | **1604** | **-2.058%** | 1612 | -1.569% | 1619 | -1.142% |
| X-n106-k14 | 26362 | 27162 | 3.035% | 27069 | 2.682% | 26966 | 2.291% | **26864** | **1.904%** | R102 | 1467 | 1567 | 6.846% | 1572 | 7.187% | **1553** | **5.891%** |
| X-n110-k13 | 14971 | 15314 | 2.291% | 15425 | 3.033% | 15386 | 2.772% | **15185** | **1.429%** | R103 | 1209 | 1493 | 23.521% | 1458 | 20.625% | **1444** | **19.467%** |
| X-n115-k10 | 12747 | 13338 | 4.636% | **13143** | **3.107%** | 13334 | 4.605% | 13162 | 3.256% | R104 | 972 | 1315 | 35.358% | 1325 | 36.387% | **1308** | **34.637%** |
| X-n120-k6 | 13332 | 13765 | 3.248% | 13741 | 3.068% | 13852 | 3.900% | **13677** | **2.588%** | R105 | 1355 | 1456 | 7.430% | 1463 | 7.947% | **1443** | **6.471%** |
| X-n125-k30 | 55539 | 58525 | 5.376% | 57943 | 4.328% | 57671 | 3.839% | **57442** | **3.426%** | R106 | 1235 | 1455 | 17.852% | 1420 | 15.017% | **1405** | **13.802%** |
| X-n129-k18 | 28940 | 29598 | 2.274% | 29517 | 1.994% | 29599 | 2.277% | **29458** | **1.790%** | R107 | 1065 | 1388 | 30.378% | 1378 | 29.438% | **1327** | **24.648%** |
| X-n134-k13 | 10916 | 11585 | 6.129% | 11468 | 5.057% | **11464** | **5.020%** | 11464 | 5.020% | R108 | 932 | 1310 | 40.543% | 1285 | 37.861% | **1266** | **35.822%** |
| X-n139-k10 | 13590 | **13812** | **1.634%** | 13863 | 2.009% | 13902 | 2.296% | 13852 | 1.928% | R109 | 1147 | 1601 | 39.594% | 1394 | 21.545% | **1359** | **18.493%** |
| X-n143-k7 | 15700 | 16257 | 3.548% | 16233 | 3.395% | **15985** | **1.815%** | 15985 | 1.815% | R110 | 1068 | 1527 | 42.978% | 1384 | 29.588% | **1282** | **20.037%** |
| X-n148-k46 | 43448 | 45036 | 3.655% | 45395 | 4.481% | 45324 | 4.318% | **44953** | **3.464%** | R111 | 1049 | 1473 | 40.460% | 1424 | 35.787% | **1366** | **30.257%** |
| X-n153-k22 | 21220 | 23478 | 10.641% | **22815** | **7.516%** | 23245 | 9.543% | 23172 | 9.199% | R112 | 949 | 1357 | 43.053% | 1278 | 34.725% | **1210** | **27.556%** |
| X-n157-k13 | 16876 | 17339 | 2.744% | 17225 | 2.068% | 17184 | 1.825% | **17131** | **1.511%** | RC101 | 1620 | 1666 | 2.852% | 1663 | 2.667% | **1661** | **2.544%** |
| X-n162-k11 | 14138 | 14664 | 3.720% | **14584** | **3.155%** | 14702 | 3.989% | 14672 | 3.777% | RC102 | 1457 | 1731 | 18.773% | 1717 | 17.813% | **1646** | **12.941%** |
| X-n167-k10 | 20557 | 21435 | 4.271% | 21305 | 3.639% | 20987 | 2.092% | **20934** | **1.834%** | RC103 | 1258 | 1760 | 39.905% | 1656 | 31.638% | **1624** | **29.094%** |
| X-n172-k51 | 45607 | 48129 | 5.530% | 47727 | 4.648% | 48252 | 5.800% | **47836** | **4.887%** | RC104 | 1132 | 1610 | 42.188% | **1497** | **32.209%** | 1524 | 34.593% |
| X-n176-k26 | 47812 | 51400 | 7.504% | 52177 | 9.130% | 51485 | 7.682% | **51164** | **7.011%** | RC105 | 1514 | 1867 | 23.340% | 1755 | 15.941% | **1751** | **15.677%** |
| X-n181-k23 | 25569 | 26097 | 2.065% | 26228 | 2.577% | 26180 | 2.390% | **26075** | **1.979%** | RC106 | 1373 | 1664 | 21.221% | 1634 | 19.035% | **1621** | **18.088%** |
| X-n186-k15 | 24145 | 25140 | 4.121% | **24909** | **3.164%** | 25046 | 3.732% | 25002 | 3.549% | RC107 | 1208 | 1683 | 39.344% | 1601 | 32.555% | **1498** | **24.027%** |
| X-n190-k8 | 16980 | 17892 | 5.371% | 17726 | 4.393% | **17547** | **3.339%** | 17547 | 3.339% | RC108 | 1114 | 1768 | 58.679% | 1564 | 40.370% | **1504** | **34.985%** |
| X-n195-k51 | 44225 | 47390 | 7.157% | 46585 | 5.336% | 46621 | 5.418% | **46121** | **4.287%** | RC201 | 1262 | 1577 | 24.980% | 1606 | 27.278% | **1533** | **21.493%** |
| X-n200-k36 | 58578 | 61199 | 4.474% | **61048** | **4.217%** | 61388 | 4.797% | 61388 | 4.797% | RC202 | 1092 | 1553 | 42.177% | 1480 | 35.494% | **1433** | **31.191%** |
| X-n209-k16 | 30656 | **31876** | **3.980%** | 32005 | 4.400% | 32334 | 5.474% | 32216 | 5.089% | RC203 | 924 | 1465 | 58.601% | 1490 | 61.308% | **1439** | **55.787%** |
| X-n228-k23 | 25742 | 28798 | 11.872% | 28328 | 10.046% | **27641** | **7.377%** | 27641 | 7.377% | RC204 | 784 | 1372 | 75.112% | 1278 | 63.114% | **1225** | **56.350%** |
| X-n237-k14 | 27042 | **29595** | **9.441%** | 29830 | 10.310% | 29816 | 10.258% | 29816 | 10.258% | RC206 | 1051 | 1573 | 49.653% | **1447** | **37.665%** | 1456 | 38.522% |
| X-n247-k50 | 37274 | 40639 | 9.028% | **40456** | **8.537%** | 41266 | 10.710% | 41266 | 10.710% | RC207 | 963 | 1694 | 75.927% | 1503 | 56.091% | **1433** | **48.821%** |
| X-n251-k28 | 38684 | 40399 | 4.433% | **40360** | **4.333%** | 40725 | 5.276% | 40505 | 4.707% | RC208 | 776 | 1465 | 88.764% | 1433 | 84.641% | **1335** | **72.014%** |
| Avg. Gap | | 5.096% | | 4.625% | | 4.707% | | **4.261%** | | Avg. Gap | | 36.573% | | 30.828% | | **27.114%** | |
| # Best (Best/Total) | | 3/27 | | 8/27 | | 4/27 | | 12/27 | | # Best (Best/Total) | | 1/27 | | 2/27 | | 24/27 | |

(*i.e.*, the version without RGCR and TSNR). Results in Fig. 6(c) show that, except for CVRP, CCL consistently reduces the average length compared to the baseline. This demonstrates that CCL is a plug-and-play strategy, which can be integrated into VRP solvers with various input structures.

## C.7 EVALUATION ON REAL-WORLD BENCHMARK INSTANCES

Table 13 and Table 14 present per-instance results on small and large-scale real-world benchmarks, respectively. The small-scale set consists of 27 CVRP and 27 VRPTW instances, while the large-scale set contains 60 VRPTW instances. Following (Zhou et al., 2024a), the model is trained on 16 synthetic tasks with $N = 100$ and directly applied to all instances in a zero-shot manner. For CVRP, we set the test-time update probability $P_{ts}$ to 0.1, denoted as CCL. We also design an ensemble variant, CCL-Ens, which applies the trained model with four update probabilities $P_{ts} \in \{0.1, 0.15, 0.25, 0.3\}$ and selects the best solution. For small-scale VRPTW, $P_{ts}$ is fixed at 0.25.

On the CVRP benchmark, CCL yields a slightly higher average gap compared to CaDA. However, its ensemble variant CCL-Ens achieves the best overall performance, demonstrating the benefit of test-time adaptation via varying update probabilities. In total, CCL and CCL-Ens together outperform baselines on 16 out of 27 instances. On the VRPTW benchmark, CCL attains the lowest average gap and surpasses baselines on 24 out of 27 small-scale instances and 35 out of 60 large-scale instances.

These results indicate that our method can be effectively deployed in real-world settings, especially on complex constraints such as time windows.

Table 14: Per-instance results on large-scale real-world instances. The customer number is 600.

| VRPTW Instances | Opt. | RF-TE Obj. ↓ | Gap ↓ | Time ↓ | CaDA Obj. ↓ | Gap ↓ | Time ↓ | CCL Obj. ↓ | Gap ↓ | Time ↓ |
|---|---|---|---|---|---|---|---|---|---|---|
| C1-6-1 | 14077 | 17537 | 24.583% | 1.4s | 17355 | 23.290% | 1.5s | 16418 | **16.633%** | 2.5s |
| C1-6-10 | 13618 | 35201 | 158.498% | 1.2s | 24365 | 78.924% | 1.3s | 17861 | **31.162%** | 1.5s |
| C1-6-2 | 13948 | 20505 | **47.007%** | 1.1s | 21571 | 54.650% | 1.1s | 20980 | 50.413% | 1.6s |
| C1-6-3 | 13757 | 22648 | **64.635%** | 1.1s | 23142 | 68.226% | 1.1s | 24090 | 75.117% | 1.6s |
| C1-6-4 | 13539 | 22743 | 67.986% | 1.1s | 22601 | **66.937%** | 1.1s | 23787 | 75.698% | 1.7s |
| C1-6-5 | 14067 | 18265 | 29.845% | 1.2s | 17643 | 25.423% | 1.1s | 16368 | **16.359%** | 1.6s |
| C1-6-6 | 14071 | 22405 | 59.229% | 3.3s | 19998 | 42.123% | 3.3s | 19481 | **38.449%** | 4.9s |
| C1-6-7 | 14067 | 26369 | 87.456% | 1.3s | 23920 | 70.046% | 1.2s | 16848 | **19.771%** | 1.6s |
| C1-6-8 | 13991 | 26618 | 90.248% | 1.2s | 20504 | 46.549% | 1.1s | 16844 | **20.390%** | 1.6s |
| C1-6-9 | 13665 | 60014 | 339.196% | 1.5s | 36753 | 168.967% | 1.3s | 18274 | **33.733%** | 1.6s |
| C2-6-1 | 7752 | 15193 | 95.983% | 3.1s | 12018 | **55.027%** | 3.1s | 12410 | 60.084% | 4.9s |
| C2-6-10 | 7124 | 29321 | 311.586% | 1.3s | 17609 | 147.182% | 1.1s | 12161 | **70.707%** | 1.6s |
| C2-6-2 | 7472 | 14789 | 97.939% | 3.1s | 14420 | 93.000% | 3.1s | 13642 | **82.587%** | 4.7s |
| C2-6-3 | 7215 | 15033 | **108.358%** | 3.1s | 16259 | 125.350% | 3.1s | 18351 | 154.345% | 4.7s |
| C2-6-4 | 6877 | 15039 | **118.685%** | 3.1s | 16236 | 136.091% | 3.1s | 18952 | 175.585% | 4.8s |
| C2-6-5 | 7554 | 22865 | 202.695% | 1.2s | 13343 | 76.640% | 1.1s | 13040 | **72.628%** | 1.5s |
| C2-6-6 | 7450 | 22171 | 197.605% | 1.2s | 13312 | 78.689% | 1.1s | 12032 | **61.508%** | 1.5s |
| C2-6-7 | 7491 | 25219 | 236.644% | 3.1s | 14632 | 95.320% | 3.1s | 12908 | **72.307%** | 4.8s |
| C2-6-8 | 7304 | 22619 | 209.692% | 1.2s | 13469 | 84.413% | 1.1s | 12266 | **67.942%** | 1.5s |
| C2-6-9 | 7303 | 23663 | 224.009% | 3.1s | 15017 | 105.622% | 3.1s | 12569 | **72.103%** | 4.7s |
| R1-6-1 | 21274 | 29154 | 37.039% | 1.2s | 25041 | **17.706%** | 1.1s | 26556 | 24.827% | 1.6s |
| R1-6-10 | 17584 | 30508 | 73.502% | 1.2s | 26126 | 48.581% | 1.1s | 26121 | **48.552%** | 1.6s |
| R1-6-2 | 18520 | 26017 | **40.482%** | 1.1s | 26262 | 41.805% | 1.1s | 27011 | 45.849% | 1.7s |
| R1-6-3 | 16875 | 26105 | **54.697%** | 1.1s | 26601 | 57.636% | 1.1s | 28093 | 66.478% | 1.6s |
| R1-6-4 | 15721 | 24450 | 55.526% | 1.1s | 24521 | 55.978% | 1.1s | 26939 | 71.359% | 1.9s |
| R1-6-5 | 19295 | 31715 | 64.370% | 1.2s | 24975 | **29.438%** | 1.1s | 25367 | 31.470% | 1.8s |
| R1-6-6 | 17764 | 25692 | 44.632% | 1.1s | 25559 | **43.883%** | 1.1s | 26890 | 51.376% | 1.6s |
| R1-6-7 | 16496 | 25749 | **56.090%** | 1.1s | 26401 | 60.043% | 1.1s | 26693 | 61.813% | 1.6s |
| R1-6-8 | 15584 | 23857 | **53.084%** | 1.1s | 24533 | 57.421% | 1.1s | 24796 | 59.109% | 1.5s |
| R1-6-9 | 18474 | 30700 | 66.179% | 1.2s | 25067 | **35.687%** | 1.1s | 25931 | 40.364% | 1.7s |
| R2-6-1 | 15145 | 31072 | 105.159% | 1.2s | 22482 | 48.442% | 1.1s | 21855 | **44.302%** | 1.6s |
| R2-6-10 | 11837 | 35862 | 202.965% | 1.2s | 23395 | 97.643% | 1.1s | 19894 | **68.066%** | 1.6s |
| R2-6-2 | 12976 | 22676 | 74.749% | 1.0s | 21124 | **62.789%** | 1.0s | 24214 | 86.602% | 1.6s |
| R2-6-3 | 10455 | 20072 | 91.979% | 1.0s | 19274 | **84.347%** | 1.0s | 24258 | 132.016% | 1.8s |
| R2-6-4 | 7915 | 16925 | 113.848% | 1.0s | 16589 | **109.603%** | 1.0s | 22399 | 183.012% | 1.7s |
| R2-6-5 | 13790 | 33895 | 145.790% | 1.2s | 22311 | 61.789% | 1.1s | 21581 | **56.495%** | 1.7s |
| R2-6-6 | 11848 | 22695 | 91.555% | 1.1s | 20914 | **76.522%** | 1.0s | 22125 | 86.744% | 1.5s |
| R2-6-7 | 9770 | 19723 | 101.867% | 1.0s | 18900 | **93.443%** | 1.0s | 23231 | 137.772% | 1.8s |
| R2-6-8 | 7512 | 16596 | **120.918%** | 1.0s | 16716 | 122.515% | 1.0s | 23249 | 209.479% | 2.0s |
| R2-6-9 | 12737 | 35883 | 181.727% | 1.3s | 22963 | 80.289% | 1.1s | 20863 | **63.801%** | 1.7s |
| RC1-6-1 | 16944 | 44173 | 160.697% | 1.3s | 28659 | 69.138% | 1.2s | 22506 | **32.824%** | 1.7s |
| RC1-6-10 | 15651 | 47967 | 206.473% | 1.3s | 33429 | 113.586% | 1.2s | 23311 | **48.940%** | 1.6s |
| RC1-6-2 | 15891 | 24480 | 54.053% | 1.1s | 25282 | 59.100% | 1.1s | 23526 | **48.050%** | 1.6s |
| RC1-6-3 | 15181 | 23667 | 55.896% | 1.1s | 23640 | **55.718%** | 1.1s | 23809 | 56.831% | 1.6s |
| RC1-6-4 | 14753 | 23076 | 56.414% | 1.1s | 22451 | 52.177% | 1.1s | 22178 | **50.327%** | 1.6s |
| RC1-6-5 | 16536 | 45720 | 176.483% | 1.3s | 29036 | 75.589% | 1.2s | 22505 | **36.095%** | 1.7s |
| RC1-6-6 | 16473 | 47520 | 188.467% | 1.3s | 30741 | 86.611% | 1.2s | 22586 | **37.107%** | 1.7s |
| RC1-6-7 | 16055 | 45359 | 182.517% | 1.3s | 31323 | 95.094% | 1.2s | 21936 | **36.628%** | 1.6s |
| RC1-6-8 | 15892 | 42548 | 167.736% | 1.3s | 32086 | 101.903% | 1.2s | 22792 | **43.420%** | 1.6s |
| RC1-6-9 | 15804 | 47394 | 199.896% | 1.3s | 33968 | 114.940% | 1.2s | 24490 | **54.966%** | 1.7s |
| RC2-6-1 | 11966 | 46194 | 286.041% | 1.3s | 26852 | 124.401% | 1.2s | 20004 | **67.172%** | 1.6s |
| RC2-6-10 | 8973 | 44372 | 394.489% | 1.3s | 28880 | 221.844% | 1.2s | 17107 | **90.643%** | 1.6s |
| RC2-6-2 | 10337 | 21343 | 106.474% | 1.1s | 20623 | **99.509%** | 1.1s | 21034 | 103.485% | 1.6s |
| RC2-6-3 | 8895 | 17942 | 101.711% | 1.0s | 17270 | **94.156%** | 1.0s | 22858 | 156.979% | 1.6s |
| RC2-6-4 | 6968 | 14864 | 113.333% | 1.0s | 14459 | **107.521%** | 1.0s | 18521 | 165.820% | 1.6s |
| RC2-6-5 | 11081 | 45546 | 311.039% | 1.3s | 27674 | 149.750% | 1.2s | 19270 | **73.906%** | 1.6s |
| RC2-6-6 | 10831 | 46487 | 329.223% | 1.3s | 27920 | 157.790% | 1.2s | 18374 | **69.651%** | 1.6s |
| RC2-6-7 | 10289 | 46929 | 356.091% | 1.3s | 28173 | 173.806% | 1.2s | 18364 | **78.475%** | 1.6s |
| RC2-6-8 | 9779 | 45416 | 364.424% | 1.3s | 29578 | 202.464% | 1.2s | 18548 | **89.672%** | 1.7s |
| RC2-6-9 | 9436 | 44922 | 376.070% | 1.3s | 29079 | 208.171% | 1.2s | 16951 | **79.642%** | 1.7s |
| Avg. | 12694 | 29558 | 145.593% | 1.4s | 22917 | 88.188% | 1.4s | 20634 | **70.961%** | 2.0s |
| # Best (Best/Total) | | | 10/60 | | | 15/60 | | | 35/60 | |

