# OpenReview forum: "Chain-of-Context Learning: Dynamic Constraint Understanding for Multi-Task VRPs"
_ICLR.cc/2026/Conference — ICLR 2026 Poster_

### Official Review · Reviewer_xK8d · 2025-10-31

**Soundness:** 2
**Presentation:** 3
**Contribution:** 2
**Rating:** 6
**Confidence:** 4

**Summary:**

The paper proposes Chain-of-Context Learning (CCL), a new neural framework for multi-task Vehicle Routing Problems (VRPs) that aims to improve constraint awareness and dynamic adaptation during solution construction. CCL combines two novel modules: Relevance-Guided Context Reformulation (RGCR), which adaptively highlights salient constraints for the current state, and Trajectory-Shared Node Re-embedding (TSNR), which stepwise updates node embeddings based on historical context and multi-trajectory exploration. The model achieves better results on multi-task VRP benchmarks of 48 variants.

**Strengths:**

1. This paper is clearly written.
2. CCL achieves state-of-the-art results on all in-distribution tasks and most out-of-distribution tasks, with performance improvements that are robust to hyperparameter and random seed variations, as shown in standard deviation analyses.
3. The ablation study is thorough.

**Weaknesses:**

1. **My main concern with this paper is that there is no theoretical basis for your design.**


Some designs are not reasonable, violating some common recognition. You mentioned in the Introduction that "They fail to leverage information accumulated in previous decoding steps, thereby limiting the model’s ability to capture and preserve historical preferences, which is an essential factor for coherent sequential decision-making." However, usually we define the RL process on the Markov Process, but your statement violates the Markov property. If you intend to include all past information in the state space, this would undoubtedly significantly increase the learning complexity, but this phenomenon is not observed in the training curves.

Specifically, I do not get the importance of providing the whole change trace of constraints as you mentioned. Your example "For example, if a customer’s time window is about to be violated, the model should prioritize handling that node immediately.", why can the customer feel the emergency with only existing scaler-based methods? They will probably make a wise decision based on the low number.

We know that, according to the Markov property, we can simplify this problem by partially solving it. According to you, such simplification (losing information about historical constraints) is actually detrimental to decision-making, which clearly contradicts the facts.

In conclusion, I believe the motivation in this paper is somehow flawed, and the authors should rethink their interpretation of their empirical findings.

**Questions:**

Given that many of your settings are flawed yet yield effective results, I would appreciate it if you could provide a reasonable explanation. If you can articulate the strengths in a reasonable manner, I am willing to increase the rating.

---

> ### Author Response · Authors · 2025-11-27
>
> Thank you very much for your time and effort in reviewing our work. We are glad to know that you find our paper is clearly written, and the ablation study is thorough. We address your concerns as follows.
>
>
> > **W1: Markov Property.**
>
> We apologize for the inaccurate statement. We **do not intend to include the full history of all past steps**. Instead, at each step, we use the node embeddings from the previous step to make the current decision.
>
> In the MDP formulation of VRP, the current state consists of two components: the context (e.g., remaining capacity, current time) and node embeddings. Existing multi-task VRP solvers update only the context at each step, while node embeddings remain fixed at their initial values throughout decoding. **This misaligned context–node pair causes inaccurate state estimation**, which leads the policy to misjudge node priority, ultimately producing suboptimal routing decisions.
>
> To address this, our method re-embeds candidate nodes at each step, updating both the node embeddings and context attributes within the environment. This **ensures that the node representation remains aligned while preserving the Markov property**.
>
> > **W2: Customer Emergency.**
>
> We thank the reviewer for the question. In our model, each node embedding contains its own time-window and demand information, while the context embedding encodes the current time and used capacity. In TSNR, the node embedding serves as the query and the context embedding serves as the key/value. This interaction **allows the node to learn a relational representation between its time window and the current time**, which helps the model estimate how urgent the node is at that step.
>
> > **Q1: Reasons for Performance Improvement.**
>
> Apologies for the confusing statements. We have revised them in Section 1. Specifically, our improvement arise from three main factors:
>
> **(1) Context–Node Alignment in RL State.** We treat context and node status as a pair in the state space, ensuring both reflect the current decoding step. During environment updates, both are updated simultaneously, unlike prior methods that update only context.
>
> **(2) Adaptive Constraint Prioritization (RGCR).** In multi-constraint scenarios, RGCR automatically learns the relative importance of constraints at each step. This enables the model to focus on the most critical or urgent ones.
>
> **(3) Efficient Multi-Trajectory Node Re-embedding (TSNR).** To handle multi-trajectory exploration efficiently, TSNR learns a shared node status across trajectories. This significantly reducing computational cost while ensuring each node can adapt to the current context.
>
> For a clear comparison, we provide an MDP formulation of the existing method here.
>
> The multi-task VRP solver acts as a single agent, using the encoder-decoder architecture as its policy network. The policy generates a node sequence autoregressively, using a Markov Decision Process (MDP) environment
> \begin{equation}
>     \mathcal{M} = (\mathcal{S}, \mathcal{A}, \mathcal{P}, \mathcal{R}).
> \end{equation}
> **State ($\mathcal{S}$)** consists of node embeddings and context embeddings. During decoding, following [1–2][7-8], the model explores from diverse starting points, forming multiple trajectories in parallel. Each trajectory maintains its own context (e.g., current time and used capacity), while all trajectories share the same set of node embeddings.
>
> **Action ($\mathcal{A}$)** corresponds to selecting the next node to visit. The policy network takes the current state as input and generates a trajectory-specific probability distribution over feasible nodes, allowing each trajectory to independently select its next action based on the predicted probabilities.
>
> **Transition ($\mathcal{P}$)** updates the environment after a node is selected. This modifies the environmental routing information, such as the vehicle’s current position and remaining capacity. The updated environment then defines the next context embedding and continues the decision process.
>
> **Reward ($\mathcal{R}$)** is defined as the negative total route length. After all nodes are visited, each trajectory computes its own negative route length as the reward. These rewards, together with the action log-probabilities produced by the policy network, are aggregated to form a single training objective. The policy network parameters $\theta$ are then updated using the REINFORCE gradient [1]:
> \begin{equation}
> \nabla_\theta J(\theta) = \frac{1}{N} \sum_{i=1}^{N} (R_i - b) \nabla_\theta \log \pi_\theta(a_i \mid s_i),
> \end{equation}
> where $i$ is the index of the trajectory and $\pi_\theta(a_i \mid s_i)$ denotes the probability assigned to action $a_i$ conditioned on state $s_i$. $b$ is a shared baseline used to reduce gradient variance, computed as the average reward over all trajectories.
>
> ## References
> [1] Simple statistical gradient-following algorithms for connectionist reinforcement learning. In Machine Learning, 1992.

---

### Official Review · Reviewer_Uhy6 · 2025-10-31

**Soundness:** 3
**Presentation:** 3
**Contribution:** 2
**Rating:** 6
**Confidence:** 4

**Summary:**

This paper addresses the challenge of solving multi-task VRPs, where existing neural solvers often fail to react to dynamic, step-wise changes in constraints and node states. The authors propose Chain-of-Context Learning (CCL), a novel framework that enhances the decoding process. CCL introduces two key modules: (1) a Relevance-Guided Context Reformulation (RGCR) module, which constructs a step-wise context by adaptively prioritizing salient constraint information, and (2) a Trajectory-Shared Node Re-embedding (TSNR) module, which efficiently updates node representations based on this dynamic context, capturing sequential dependencies without full re-computation. The authors evaluate CCL on a comprehensive benchmark of 48 VRP variants, demonstrating SOTA performance on all 16 in-distribution tasks and 32 out-of-distribution (OOD) tasks.

**Strengths:**

1. The core contribution, explicitly modeling step-wise constraint dynamics within the decoder, is a novel and intuitive approach.
2. The paper is well-written, and its structure is logical.
3. The paper's empirical evaluation is thorough and rigorous. The use of a large benchmark (48 variants)  provides strong evidence for the method's generalization capabilities. The ablation studies are comprehensive.

**Weaknesses:**

1. In the Introduction (lines 56–57), the paper claims that existing methods fail to consider node priority. Could the authors provide illustrative examples or visualizations to demonstrate this point and further clarify how the proposed method captures node priority more effectively?

2. Works such as Jieyi Bi’s PIP and others have also addressed constraint-aware learning, though often in single-task settings like TSPTW. It would be valuable to distinguish the proposed method from these studies and discuss whether such single-task methods could be adapted to multi-task problems.

3. The rationale for the superiority of RGCR in integrating constraints into the embedding space is unclear. Incorporating constraints to improve performance is intuitively reasonable, but the paper does not explain why the proposed approach achieves better integration. A more intuitive explanation or analysis would be helpful.

**Questions:**

In Table 3, the ablation studies are conducted based on ReLD (CCL$^\dagger$}). I would like to know whether the effectiveness of the proposed model depends on the ReLD module.

---

> ### Author Response · Authors · 2025-11-27
>
> Thank you very much for your time and effort in reviewing our work. We are glad to know that you find our paper well-written, the method is novel, and it presents a thorough evaluation. We address your concerns as follows.
>
> > **W1: Capturing Node Priority More Effectively**
>
> We acknowledge that our original statement was imprecise. What we meant is that the priority of nodes changes across decoding steps. Existing solvers rely on static node embeddings, which remain fixed from the first to the last step and thus cannot reflect this dynamic property.
>
> Our method proposes to re-embed candidate nodes at each decoding step. Specifically, node embeddings contain the node-specific attributes, such as time window, while the context embedding encodes the current context status, e.g., the current time. By combining these embeddings, the model can learn the relationship between a node’s time window and the current time. This allows the model to assess the node urgency and dynamically determine node priorities at the current decoding step.
>
>
> > **W2: Similar Work on Constraint-aware Learning.**
>
> We thank the reviewer for raising this point. Our constraint-aware learning aims to find the constraint importance level. For example, at a decoding step, we need to pay more attention to the time window or distance limit. In contrast, PIP [1] aims to prevent constraint violations in TSPTW or TSPDL by learning feasibility masks. For example, when solving TSPTW, it may happen that the current time exceeds all node end times while some nodes remain unvisited. Following RouteFinder [2], CaDA [3], MTPOMO [4], and MvMoE [5], our vehicle returns to the depot and resets the context (e.g., current time and used capacity are set to zero) in this scenario. In the new sub-route, the previously unvisited nodes become feasible again and can be visited.
>
> While the constraint violation is a valuable line of research, it is orthogonal to our setting and not part of the standard benchmark protocol we follow. Moreover, PIP focuses on single-task training (e.g., TSPTW or TSPDL). Extending it to handle multiple tasks with different combinations of constraints (e.g., a TSP with both TW and DL or other constraints) may substantially increase the model complexity. However, in future work, we will investigate how to incorporate PIP's feasibility estimator into CCL, thereby achieving more efficient constraint understanding.
>
> > **W3: Superiority of RGCR.**
>
> We thank the reviewer for the comment. Our key idea is to **strengthen the influence of important constraints rather than treating all constraints equally**. In RGCR, we compute the correlation between the current node embedding and each constraint embedding. Since the current node embedding reflects the last action’s preference, the resulting correlation scores provide guidance on which constraints the policy should emphasize at the current step.
>
> To validate this, we test 16 in-domain VRP variants, each with 1,000 instances, and compare RGCR with three alternatives: (1) Baseline, which directly concatenates the constraint attributes; (2) Embedding Concatenation, which embeds each constraint into a high-dimensional space and concatenates them; and (3) Random Weights, using random importance weights to replace our correlation scores.
>
> We present both the model complexity and performance results. Specifically, "# Params" denotes the total number of parameters in the encoder and decoder, and "Time" is the accumulated inference time over 1,000 instances. "Avg.Gap" denotes the average gap across all 16 tasks, while "Avg.Gap w/ TW" refers to the 8 tasks that have time window constraints.
>
> | Method         | # Params (M)  | Time (s) | Avg.Gap | Avg.Gap w/ TW |
> |----------------|-------------|------|---------|----------------|
> | Baseline       | 3.39+0.56   | 5.2  | 1.156%   | 0.729%          |
> | Embedding Concatenation  | 3.39+0.66   | 5.7  | 1.141%   | 0.713%          |
> | Random Weights | 3.39+0.66   | 6.5  | 1.092%   | 0.707%          |
> | RGCR           | 3.39+0.66   | 6.5  | 1.100%   | 0.689%          |
>
>
> Compared with the baseline, RGCR achieves strong performance while increasing the model size by only 0.1 M parameters and adding 1.3 s to inference time. Compared to random weights, RGCR shows modest performance in average gap across the 16 tasks, but demonstrates clear superiority on tasks with time windows. These results demonstrate that RGCR benefits more on complex tasks than on simpler ones. This may be attributed to the fact that tasks without time windows often include a lot of padding information, which may introduce some noise during model training.
>
> In the revised manuscript, we have included the core results in Section 4.4, with the corresponding experimental details presented in Appendix C.3.2.

---

> ### Author Response · Authors · 2025-11-27
>
> > **Q1: Ablation Results without ReLD.**
>
> We conducted an ablation study to examine whether the effectiveness of the proposed model depends on the ReLD module. Starting from CCL, we remove RGCR, TSNR, and both components. The corresponding results are shown below:
> | Methods      | Obj.   | Gap    | Time |
> |--------------|--------|--------|------|
> | CCL†         | 11.447 | 1.10%  | 6.5s |
> | CCL          | 11.464 | 1.28%  | 5.4s |
> | - RGCR        | 11.477 | 1.40%  | 4.6s |
> | - TSNR        | 11.518 | 1.76%  | 2.8s |
> | - TSNR - RGCR   | 11.529 | 1.88%  | 2.0s |
>
> The results show that both RGCR and TSNR remain effective when added to CCL. Specifically, CCL reduces the average gap by 0.6% compared with CCL-TSNR-RGCR, whereas CCL† provides only a 0.18% improvement over CCL. This indicates that the main performance improvement comes from CCL rather than ReLD. We have included this experiment in Appendix C.3.1.
>
> ## References
> [1] Learning to Handle Complex Constraints  for Vehicle Routing Problems. In NeurIPS, 2024.
>
> [2] Routefinder: Towards foundation models for vehicle routing problems. In TMLR, 2025.
>
> [3] Cada: Cross-problem routing solver with constraint-aware dual-attention. In ICML, 2025.
>
> [4] Multi-task learning for routing problem with cross-problem zero-shot generalization. In SIGKDD, 2024.
>
> [5] MVMoe: Multi-task vehicle routing solver with mixture-of-experts*. In ICML, 2024.

---

### Official Review · Reviewer_yziX · 2025-11-07

**Soundness:** 3
**Presentation:** 2
**Contribution:** 2
**Rating:** 6
**Confidence:** 3

**Summary:**

This paper proposes Chain-of-Context Learning (CCL), a novel reinforcement learning framework for solving multi-task Vehicle Routing Problems (VRPs). CCL introduces two key modules:  Relevance-Guided Context Reformulation (RGCR), which dynamically prioritizes constraints at each decoding step. Trajectory-Shared Node Re-embedding (TSNR), which efficiently refines node embeddings across multiple trajectories using shared context. The method is evaluated on 48 VRP variants (16 in-distribution, 32 out-of-distribution), demonstrating state-of-the-art performance on both seen and unseen tasks, particularly those involving complex constraints like time windows.

**Strengths:**

**Originality**: The idea of step-wise, context-aware node re-embedding is novel in the multi-task VRP setting. Unlike prior methods that use static embeddings, CCL captures evolving constraint priorities and node states, addressing a clear gap in the literature.

**Quality**: The paper is technically sound, with well-designed modules (RGCR and TSNR) and thorough experiments. Ablation studies and complexity analyses validate the design choices.

**Weaknesses:**

**Weaknesses**

1. **Methodological Complexity and Limited Generalizability**: The proposed CCL framework introduces significant architectural complexity through its two specialized modules (RGCR and TSNR). While effective for multi-task VRPs, the approach appears highly tailored to this specific problem domain. The paper would benefit from discussing how these components might generalize to other combinatorial optimization problems beyond VRPs, or what adaptations would be necessary for broader applicability.
2. **Narrow Technical Contribution Focus**: The primary innovation appears concentrated on neural architecture design rather than broader methodological contributions. The work introduces novel network components but doesn't substantially advance the underlying reinforcement learning paradigm, problem formulations, or theoretical foundations. A more balanced contribution spanning architectural, algorithmic, and theoretical dimensions would strengthen the paper's impact.
3. **Insufficient Formalization of RL Framework**: The reinforcement learning formulation lacks rigorous mathematical specification. The paper would be significantly strengthened by explicitly defining the core Markov Decision Process (MDP) components.
4. **Presentation Issues in Figures**:
    - **Figure 1**: The loss term contains unclear symbols or potential encoding issues.
    - **Figure 2**: There is a typo in the label "Bais" which should be corrected to "Bias".

**Questions:**

1. The update probabilities $P_{tr}$ and $P_{ts}$ are set empirically. Have the authors considered a learned or adaptive scheduling mechanism for these parameters?
2. How does CCL perform on highly constrained or infeasible instances? Is there a mechanism to handle or recover from constraint violations during decoding?
3. The paper focuses on Euclidean VRPs. Can CCL be extended to non-Euclidean or graph-structured routing problems? What modifications would be required?

---

> ### Author Response · Authors · 2025-11-27
>
> Thank you very much for your time and effort in reviewing our work. We are glad to know that you find our paper is technically sound, and the method is novel for multi-task VRP. We address your concerns as follows.
>
> > **W1: Complexity and Generalize beyond VRPs.**
>
> We thank the reviewer for the comment. Same as RouteFinder [2], CaDA [3], MTPOMO [4], and MvMoE [5], our work also focuses on solving routing problems solely. Please kindly note that CCL has already been evaluated on 48 routing tasks, demonstrating strong generalizability and adaptability across different scenarios. Moreover, the design of CCL can be useful for other decision-making problems. For instance, when multiple constraints coexist, RGCR can be used to determine the relative importance of each constraint.
>
> For a direct response to the comment, we conducted a preliminary experiment on the Flexible Flow Shop Problem (FFSP). When assigning an operation to a machine, we incorporated TSNR to allow operation embeddings to integrate information from the current machine, thereby updating the operation’s state.
>
> | Epoch | 1    | 3    | 5    | 7    | 9    | 10   | 20   | 30   | 40   | 50   | 60   | 70   | 80   | 90   | 100  |
> |-------|------|------|------|------|------|------|------|------|------|------|------|------|------|------|------|
> | Score | 40.3 | 39.5 | 39.3 | 39.1 | 37.8 | 37.3 | 36.5 | 37.1 | 37.2 | 36.8 | 36.2 | 36.1 | 36.0 | 35.8 | 36.2 |
> | Loss  | -0.2 | 0.8  | 0.6  | 0.6  | -1.5 | -2.7 | -3.7 | -2.6 | -2.8 | -3.2 | -3.9 | -3.7 | -3.8 | -2.7 | -2.8 |
>
>
> Our results show that the method converges, but training can become slightly unstable from the middle to late stages. This suggests that certain modifications to TSNR may be needed for the best performance, for example, to filter out irrelevant information in the machine embeddings that does not contribute to subsequent decision-making.
>
> > **W2: Methodological Contributions.**
>
> We appreciate the reviewer’s comment. While we acknowledge that the RGCR and TSNR modules are primarily architectural contributions, the paper also advances the RL framework in the context of solving multi-task VRPs:
>
> **(1) Context–Node Alignment in RL State.** We treat context and node status as a pair in the state space, ensuring both reflect the current decoding step. During environment updates, both are updated simultaneously, unlike prior methods that update only the context.
>
> **(2) Adaptive Constraint Prioritization (RGCR).** In multi-constraint scenarios, RGCR automatically learns the relative importance of constraints at each step. This enables the model to focus on the most critical or urgent ones.
>
> **(3) Efficient Multi-Trajectory Node Re-embedding (TSNR).** To handle multi-trajectory exploration efficiently, TSNR learns a shared node status across trajectories. This significantly reduces computational cost while ensuring each node can adapt to the current context.
>
> These points show that our contribution is not limited to architectural design but also includes algorithmic and methodological innovations.

---

> ### Author Response · Authors · 2025-11-27
>
> > **W3: RL Formulation.**
>
> We appreciate the reviewer’s insightful comments. We have now added the full formulation in Section 2 of the revised manuscript.
>
> The multi-task VRP solver acts as a single agent, using the encoder-decoder architecture as its policy network. The policy generates a node sequence autoregressively, using a Markov Decision Process (MDP) environment
> \begin{equation}
>     \mathcal{M} = (\mathcal{S}, \mathcal{A}, \mathcal{P}, \mathcal{R}).
> \end{equation}
> **(1) State ($\mathcal{S}$)** consists of node embeddings and context embeddings. During decoding, following [1–2][7-8], the model explores from diverse starting points, forming multiple trajectories in parallel. Each trajectory maintains its own context (e.g., current time and used capacity), while all trajectories share the same set of node embeddings.
>
> **(2) Action ($\mathcal{A}$)** corresponds to selecting the next node to visit. The policy network takes the current state as input and generates a trajectory-specific probability distribution over feasible nodes, allowing each trajectory to independently select its next action based on the predicted probabilities.
>
> **(3) Transition ($\mathcal{P}$)** updates the environment after a node is selected. This modifies the environmental routing information, such as the vehicle’s current position and remaining capacity. The updated environment then defines the next context embedding and continues the decision process.
>
> **(4) Reward ($\mathcal{R}$)** is defined as the negative total route length. After all nodes are visited, each trajectory computes its own negative route length as the reward. These rewards, together with the action log-probabilities produced by the policy network, are aggregated to form a single training objective. The policy network parameters $\theta$ are then updated using the REINFORCE gradient [1]:
> \begin{equation}
> \nabla_\theta J(\theta) = \frac{1}{N} \sum_{i=1}^{N} (R_i - b) \nabla_\theta \log \pi_\theta(a_i \mid s_i),
> \end{equation}
> where $i$ is the index of the trajectory and $\pi_\theta(a_i \mid s_i)$ denotes the probability assigned to action $a_i$ conditioned on state $s_i$. $b$ is a shared baseline used to reduce gradient variance, computed as the average reward over all trajectories.
>
>
> > **W4: Presentation Issues for Loss Notation and "bais".**
>
> We thank the reviewer for pointing this out. The typo has been corrected in both Figure 1 and Figure 2.
>
>
> > **Q1: Adaptive Update Probabilities.**
>
> We thank the reviewer for the insightful suggestion. We have experimented with a learnable scheduling mechanism by treating $P_{tr}$ as a trainable parameter initialized at 0.5. During inference, the optimized $P_{tr}$ was also used as $P_{ts}$. The results are shown below:
>
> | Method                     | Obj.    | Gap   | Time |
> |---------------------------|---------|-------|------|
> | CCL                       | 11.447  | 1.100% | 6.5s |
> | CCL + Learnable $P_{tr}$      | 11.460  | 1.216% | 5.4s |
>
> The learnable version reduces inference time but does not improve performance. We further examined the learned $P_{tr}$ values and found that they remained at the initial value of 0.5 throughout training. The reason is that the sampling process (drawing a random number and comparing it with $P_{tr}$) is not differentiable, and the action probabilities are not connected to $P_{tr}$ through the computation graph. Therefore, gradients cannot flow to this parameter, making it unlearnable under this design.
>
> For future work, we plan to estimate the magnitude of context change and let the model decide whether node embeddings should be updated, which enables a principled adaptive mechanism for TSNR.
>
> > **Q2: constraint violations.**
>
> We thank the reviewer for raising this point. Following RouteFinder[2], CaDA[3], MTPOMO[4], and MvMoE[5], our decoding process **employs the masking mechanism to distinguish feasible nodes**.
>
> Specifically, when solving the 48 tasks, the model updates context attributes such as current time and used capacity after a node is selected. The mask is then recomputed to determine the remaining feasible nodes. It may occur that the current time is later than all end times, but some nodes are not visited. Under this scenario, the vehicle will return to the depot and reset the context. **In the new sub-route, the aforementioned nodes become feasible again and can be visited**.
>
> However, in some highly constrained tasks, such as TSPTW, the number of vehicles is fixed at 1, so the vehicle cannot return to the depot until all nodes are visited. To avoid violations, recent works have proposed learning feasibility masks, such as PIP [6]. While this direction is valuable, it is orthogonal to our setting and not included in the benchmark we follow. Nonetheless, exploring such mechanisms is an interesting avenue that we plan to investigate.

---

> ### Author Response · Authors · 2025-11-27
>
> > **Q3: Graph-Structured VRPs.**
>
> Our work follows the settings of RouteFinder[2], CaDA[3], MTPOMO[4], and MvMoE[5], where each instance is represented by Euclidean coordinates. Both coordinate-based and graph-structured (distance matrix) formulations require converting raw inputs into node embeddings, and the decoding procedure remains similar. Therefore, **CCL has the potential to be adapted to existing matrix-based encoders.**
>
> We also conducted an intuitive experiment using graph-structured inputs instead of coordinates. Specifically, each node's coordinates were replaced with a vector of distances to all other nodes, which was then concatenated with demand and other attributes to form the node inputs. We retrain CCL across 16 tasks, each with 50 customers. During the 300 training epochs, we report the training loss and the validation average objective length across 128 CVRP instances (also with 50 customers). The following table shows that both the training loss and the validation scores of CCL converge quickly within the first 50 epochs, suggesting the potential of CCL for graph-structured VRPs.
>
> | Epoch | Loss   | CVRP Obj. |
> |-------|--------|-----------|
> | 0     | -0.152 | -11.654   |
> | 49    | -0.018 | -10.497   |
> | 99    | -0.013 | -10.475   |
> | 149   | -0.012 | -10.479   |
> | 199   | -0.012 | -10.473   |
> | 249   | -0.011 | -10.469   |
> | 299   | -0.010 | -10.444   |
>
>
> To further investigate the effectiveness of RGCR and TSNR within CCL, we apply this setting to retrain the corresponding baseline model (i.e., the version without RGCR and TSNR). The table below presents the comparison results. Both methods are evaluated across 16 tasks, each having 1000 instances with 50 customers. "Obj." denotes the average length across 16 in-domain tasks, and "Time" denotes the accumulated inference time over 1,000 instances.
>
> | Method                 | Obj.   | Time |
> |------------------------|--------|------|
> | Baseline          | 11.604 | 3.2s |
> | Graph-Structured CCL       | 11.538 | 6.6s |
>
> We find that CCL reduces the average length by 0.066 compared to the corresponding baseline. This demonstrates that RGCR and TSNR are plug-and-play strategies, which can be integrated into VRP solvers with various input structures.
>
> However, real-world scenarios may involve more complex inputs, such as asymmetric routing problems. These nodes may contain both row-wise and column-wise information. Extending CCL to these settings may require a mechanism to integrate that information, so that the model can capture the full asymmetric structure.
>
> ## References
> [1] Simple statistical gradient-following algorithms for connectionist reinforcement learning. In Machine Learning, 1992.
>
> [2] Routefinder: Towards foundation models for vehicle routing problems. In TMLR, 2025.
>
> [3] Cada: Cross-problem routing solver with constraint-aware dual-attention. In ICML, 2025.
>
> [4] Multi-task learning for routing problem with cross-problem zero-shot generalization. In SIGKDD, 2024.
>
> [5] MVMoe: Multi-task vehicle routing solver with mixture-of-experts*. In ICML, 2024.
>
> [6] Learning to Handle Complex Constraints  for Vehicle Routing Problems. In NeurIPS, 2024.

---

> > ### Comment · Reviewer_yziX · 2025-11-28
> >
> > I thank the authors for their detailed response and revisions. Most of my concerns have been addressed, and the formatting improvements have enhanced the paper's readability. I maintain my positive assessment.

---

> > > ### Author Response · Authors · 2025-11-28
> > >
> > > We appreciate the reviewer for acknowledging our rebuttal and keeping a positive score. We are more than happy to address the concerns if you still have any. Thanks again for your support!

---

### Official Review · Reviewer_jrJZ · 2025-11-07

**Soundness:** 2
**Presentation:** 2
**Contribution:** 2
**Rating:** 4
**Confidence:** 3

**Summary:**

This paper tackles the niche problem of multi-task vehicle routing optimisation, where a vehicle must deliver goods from a depot to customers arranged variably in 2D space as a graph of nodes. The authors introduce a method to aggregate the context (encoding information from multiple constraints and node features) while incorporating step-wise relevance of each constraint using its similarity to node features. They further add a module to incorporate sequential/temporal information from multiple trajectories and other nodes using multi-head attention. Contexts/features are encoded and actions are decoded using an encoder-decoder architecture. Results over a range of VRP tasks show slight improvements over state-of-the-art baselines in terms of performance, but with increased inference time.

**Strengths:**

* The methodology and diagram are clear
* Fairly many tasks (comparative and ablative) are tested on
* The results discussion and analysis are detailed

**Weaknesses:**

* The motivation for this architecture is not very clear to me. There have already been works that incorporate context from other nodes (Kool et al 2019) or use sequential information (Nazari et al 2018)
* The architecture consists of several components which are not novel in this space. At the same time, they are combined in a complicated way and I don’t understand the need for this complication (e.g. combining constraint embeddings with node features many times over in different ways in RGCR)
* The results show extremely incremental improvements in performance, to the point where it’s hard to tell if they came from the architecture or just the size of the networks etc.
* The degradations in inference time, by contrast, are significant
* The method only applies to a single agent (let me know if I’m wrong here as this wasn’t completely clear), unlike the SOTA baseline MTPOMO
* Minor: ‘bias’ is misspelled ‘bais’ in the architecture diagram and the bar chart

**Questions:**

* I don’t understand how the action selection is RL? It seems to be a form of offline return (sum of future rewards) maximisation but there is no value iteration or policy gradient

---

> ### Author Response · Authors · 2025-11-27
>
> Thank you very much for your time and effort in reviewing our work. We are glad to know that you find our methodology clear, and the results analysis is detailed. We address your concerns as follows.
>
> > **W1: Motivation and Similar Works.**
>
> In the MDP process, the state consists of two parts: the current context and the node embeddings. However, existing methods, e.g., **Kool et al. (2018) [1]**, keep node embeddings fixed from the first to the last decoding step, which prevents them from accurately reflecting the current state. In contrast, the proposed TSNR treats context and node as a pair in the state space, ensuring both remain consistent with the current decoding step. During environment transitions, TSNR updates them simultaneously, whereas prior methods update only the context while leaving node embeddings unchanged.
>
> While **Nazari et al. (2018) [2]** feed unvisited nodes into the network, their approach struggles in multi-constraint scenarios. To address this, we propose RGCR, which automatically learns the relative importance of different constraints at each decoding step. This enables the solver to dynamically prioritize the most critical constraints and handle complex requirements more effectively.

---

> ### Author Response · Authors · 2025-11-27
>
> > **W2&W3: Performance Gain Is Not Due to Model Size.**
>
> We thank the reviewer for the comment. Compared with the SOTA, CCL indeed increases the parameter count by 0.39M and reduces the performance gap by 0.53%. However, these improvements arise from node updates and adaptive constraint combination rather than from a larger network. Below, we justify this from three aspects.
>
> **(1) Effectiveness of RGCR.**
>
> Our key idea is to **strengthen the influence of important constraints rather than treating all constraints equally**. Although the architectural components are modest, they are necessary in the multi-constraint setting.
>
> To validate this, we test 16 in-domain VRP variants, each with 1,000 instances, and compare RGCR with three alternatives: (1) Baseline, which directly concatenates the constraint attributes; (2) Embedding Concatenation, which embeds each constraint into a high-dimensional space and concatenates them; and (3) Random Weights, using random importance weights to replace our correlation scores.
>
> We present both the model complexity and performance results. Specifically, "# Params" denotes the total number of parameters in the encoder and decoder, and "Time" is the accumulated inference time over 1,000 instances. "Avg.Gap" denotes the average gap across all 16 tasks, while "Avg.Gap w/ TW" refers to the 8 tasks that have time window constraints.
>
> |Method|# Params (M)|Time (s)|Avg.Gap|Avg.Gap w/ TW|
> |:----|----:|----:|----:|----:|
> |Baseline|3.39+0.56|5.2|1.156%|0.729%|
> |Embedding Concatenation|3.39+0.66|5.7|1.141%|0.713%|
> |Random Weights|3.39+0.66|6.5|1.092%|0.707%|
> |RGCR|3.39+0.66|6.5|1.100%|0.689%|
>
> Compared with the baseline, RGCR achieves strong performance while increasing the model size by only 0.1 M parameters and adding 1.3 s to inference time. Compared to random weights, RGCR shows modest performance in average gap across the 16 tasks, but demonstrates clear superiority on tasks with time windows. These results demonstrate that RGCR benefits more on complex tasks than on simpler ones. This may be attributed to the fact that tasks without time windows often include a lot of padding information, which may introduce some noise during model training.
>
> Moreover, **RGCR introduces no additional parameters compared to the Direct Concat setting, yet still reduces the average gap by 0.041%.** This indicates that the gains arise from improved constraint prioritization rather than model capacity. In the revised manuscript, we have included the core results in Section 4.4, with the corresponding experimental details presented in Appendix C.3.2.
>
> **(2) Effectiveness of TSNR.**
>
> **TSNR updates the re-embedded node as the input for the next decoding step. This operation introduces no additional parameters yet provides substantial improvement.** As shown in the right part of Fig. 2 (original manuscript), it reduces the gap by 0.22% and 0.15% for tasks with and without time windows, respectively. This phenomenon indicates that node status update benefits coherent sequential decision-making, especially for complex tasks with time windows.
>
> **(3) Comparison with a Heavy-Decoder Variant of SOTA.**
>
> In the original manuscript, Appendix C.4 compares CCL with a heavy-decoder variant of the SOTA model. In this variant, the number of transformer layers in the CaDA decoder is increased from 1 to 4.
>
> |Methods|Gap(%)|Memory(GiB)|Params(M)|Time(s)|
> |:----|----:|----:|----:|----:|
> |CaDA†-HD|1.53|7.55|3.37+1.0|5.1|
> |CCL|1.28|8.13|3.39+0.45|5.4|
>
> These results show that CCL achieves a comparable model cost while reducing the gap by 0.25% compared with CaDA†-HD. In addition, results in the original Appendix C.4 demonstrate that CCL consistently outperforms CaDA†-HD on tasks with time windows. These findings indicate that the effectiveness of CCL stems from its design rather than from an increased network scale. In the revised manuscript, we have added the average results in Section 5, while the additional details are maintained in Appendix C.4.

---

> ### Author Response · Authors · 2025-11-27
>
> > **W4: Inference time.**
>
> We appreciate the reviewer’s concern regarding inference speed. In RL-based frameworks, node re-embedding is a powerful but costly strategy due to multi-trajectory exploration and gradient accumulation at each decision step [3]. Our proposed Trajectory-Shared Node Re-embedding (TSNR) find a shared node status across multiple trajectories and efficiently reduces computational cost. This enables RL-based node re-embedding to be executed on a single 46 GB GPU.
>
> To further reduce the computational burden, the original manuscript thoroughly analyzed the trade-off between performance and inference time in Section 5 and Appendix C.4. Specifically, Section 5 already assessed the impact of update probabilities on model performance and inference efficiency. As shown in Fig. 3, the lightweight version of CCL† achieves an average gap of 1.38% across 16 VRPs with an average inference time of 4.6s, while the SOTA CaDA† attains a 1.63% gap in 2.7s (see Table 4). Moreover, Appendix C.4 investigated a heavy decoder variant of CaDA to evaluate performance under a similar inference budget. The results show that CCL outperforms CaDA in 10 out of 16 tasks under comparable inference time.
>
> We acknowledge the reviewer’s concern and believe that CCL still has room for speedup. In future work, we plan to improve the efficiency by designing more effcient architecture and employing accelerated Python modules.
>
>
> > **W5&Q1: RL Formulation.**
>
>
> Both our method and MT-POMO are single-agent offline RL approach.
>
> During decoding, we **explore from multiple starting points to generate multiple trajectories**. Given the current context and node status, the decoder outputs a probability distribution over feasible actions, from which each trajectory selects its next node. The environment is then updated after each action, triggering the transition to the next step. When all nodes have been visited, **the total route cost is computed as the reward**. Each trajectory maintains its own context, actions, and reward. **All trajectory rewards are then aggregated to compute a single objective** for updating the network.
>
>
> Since only a single policy network is employed and the parameters are updated based on a single objective, both our method and MT-POMO are **single-agent RL**.
>
> The environment is updated after each action, but no intermediate rewards are assigned. The total route cost is computed only after the trajectory is completed and serves as the reward. The objective function then combines this trajectory reward with the probabilities of all actions along the trajectory. Since **these probabilities originate from the network, the corresponding gradients can be computed directly from the objective function**. Therefore, this constitutes a standard MDP setup with offline RL: no intermediate rewards or value iteration are used.
>
> We apologize for the confusion. To clarify, we have revised Figure 1 to explicitly show multiple trajectories, and added the RL formulation in Section 2.
>
> > **W6: Typo of "Bias".**
>
> We thank the reviewer for pointing this out. The typo has been corrected in both Figure 1 and Figure 2.
>
> ## References
> [1] Attention, learn to solve routing problems! In ICLR, 2019.
>
> [2] Reinforcement learning for solving the vehicle routing problem. In NeurIPS, 2018.
>
> [3] Step-wise deep learning models for solving routing problems. In IEEE TII, 2020.

---

### Official Review · Reviewer_A3oC · 2025-11-08

**Soundness:** 2
**Presentation:** 3
**Contribution:** 2
**Rating:** 4
**Confidence:** 4

**Summary:**

The paper presents Chain-of-Context Learning (CCL), a novel framework designed to enhance decision-making in multi-task Vehicle Routing Problems (VRPs) by dynamically adapting to evolving constraints. Traditional heuristic methods for VRPs are computationally intensive and inflexible, while existing neural approaches rely on static embeddings that fail to capture changing constraint priorities. CCL overcomes these challenges through two key modules: the Relevance-Guided Context Reformulation (RGCR) module, which dynamically prioritizes salient constraints, and the Trajectory-Shared Node Re-embedding (TSNR) module, which refines node representations by aggregating contextual information from multiple trajectories. Using a transformer-based encoder and reinforcement learning, CCL jointly models node and constraint dynamics to make adaptive routing decisions. Extensive experiments on 48 VRP variants show that CCL consistently outperforms state-of-the-art baselines, including CaDA, achieving superior routing efficiency and generalization to out-of-distribution tasks. Ablation studies confirm the complementary effectiveness of RGCR and TSNR, while complexity analysis shows that the slight increase in inference time is justified by significant performance gains. Furthermore, CCL demonstrates robust performance on real-world VRPTW instances, achieving the lowest average performance gap among all tested methods.

**Strengths:**

1. The problem addressed in this work is of high significance in the field of operations research. It deals with a complex and practically relevant challenge that aligns with current trends in optimization and decision-making under uncertainty.

2. The authors have provided a thorough and well-balanced review of the existing literature related to the problem under study. The cited works effectively capture both the foundational research and recent advancements in the area, demonstrating a clear understanding of the current state of the art. The discussion successfully positions the proposed work within the broader research landscape, highlighting how it builds upon and differentiates itself from prior studies.

3. The manuscript is well organized and presented in a logical, easy-to-follow manner. The problem statement, methodology, and experimental results are clearly delineated, making it straightforward for the reader to follow the development of ideas. The figures, tables, and explanations are used effectively to support the narrative and to communicate key insights.

4. The authors have conducted a comprehensive experimental analysis that strengthens the credibility of their claims. The inclusion of evaluations on out-of-distribution scenarios demonstrates the robustness and generalizability of the proposed approach. Furthermore, the trade-off analysis and ablation studies provide valuable insights into the contribution of individual components and the practical implications of design choices.

**Weaknesses:**

1. The problem considered in this work is not a notoriously difficult problem to solve with existing learning and non-learning-based methods. Considering the complexities and the size of the problem, optimal solutions can be obtained by formulating the problem as Integer Linear Programming, and using commercial solvers given enough computing time. Since the problem is deterministic, computing time is not a limitation unless the problem size is significantly large. Therefore,  I would encourage the authors to provide more justification on why a learning-based method should be used for a small-scale deterministic combinatorial optimization problem.

2. The baselines used for comparison are not strong. Provided that the problem along with the six constraints do not invoke any dynamically generated tasks or uncertainties, there are many traditional methods including Integer Linear Programing (ILP) based methods to compare. I would also encourage the authors to use the LKH solver to compare the performance of their proposed method. These centralized traditional methods usually provide optimal solutions compared to learning-based method. For deterministic problems like this, where optimal solution can be pre-computed, the use of learning-based approaches is not justified. I would also encourage the authors to provide a short (1-2 sentence) description about the baselines used, including any parameter/configuration settings.

3. The manuscript lacks a formal problem formulation. In order to apply Reinforcement Learning, the problem has to be formulate as a Markov Decision Process, explaining the state space, action space, and the reward. Having the problem formulation gives the reader a better understanding of  how/why the proposed decision-making architecture works.

4. I would encourage the authors to use statistical tests such as T-test or ANOVA to compare their method against other baselines. The statistical tests can provide evidence if the difference in mean is significant.

**Questions:**

1. In equation 1, shouldn't the dimensions of H be (N+2) x D?

2. What is N&L, and C&L in figure 1?

3. What does "Multiple Trajectories" mean in figure 1?

4. What does "multi-trajectory context" mean?

5. What is the dimension of the concatenated vector in equation 5?

6. What does "trajectory" refers to in the context of the manuscript? Is it the same as a sub-route?
The linehaul and backhaul demands are a node-specific quantity. It is not clear why they are expressed using the notations for trajectory an decoding step (i and j).

---

> ### Author Response · Authors · 2025-11-27
>
> Thank you very much for your time and effort in reviewing our work. We are glad to know that you find our paper is of high significance, easy to follow, and that the experimental analysis is comprehensive. We address your concerns as follows.
>
> > **W1&W2: The Superiority of Learning-based Methods over Traditional Solvers.**
>
>
> We follow RouteFinder [1] and CaDA [2] in using HGS-PyVRP [3] as a strong traditional solver. Moreover, we compare our method against additional traditional solvers, including Gurobi [4] and LKH [5], on CVRP instances with 50 customers. The total times are accumulated over 1000 instances, which are exactly the same as the ones used in [2]. We also provide the average per-instance time for reference.
>
> The results of HGS-PyVRP are taken from [2], while the results of Gurobi and LKH are obtained using a 32-core CPU. Based on this, Gurobi further uses 4 threads per CPU core, enabling 4×32 instances to be solved in parallel. LKH is executed for 10 runs, with a 10-second time limit per instance. We set a 15-minute limit on Gurobi and report its generated (approximate) solutions.
>
> | Methods     | Obj. | Total Time | Per Instance Time|
> |-------------|:---------:|----------:|----------:|
> | HGS-PyVRP   | 10.372 | 10.4m  | 10.0s |
> | Gurobi-15min      | 10.568    | 120.2m    | 15m|
> | LKH         | 10.392    | 63s       | 1.1s|
> | CCL† (Ours)  | 10.463    | 6s        |0.2s|
>
> These results show that CCL achieves performance comparable to traditional solvers, while its total inference time is approximately 10×, 1,200×, and 100× faster than LKH, Gurobi, and HGS-PyVRP, respectively (corresponding per-instance speedups of 5×, 75×, and 50×). This demonstrates that learning-based models are practical for real-time applications, especially when solving multiple VRP instances simultaneously.
>
> Moreover, in multi-task scenarios, CCL can learn generalizable patterns across different VRP variants, without requiring experts to manually design heuristics. Once trained, the model can solve 48 VRP variants without re-training, which will broaden its practical deployment scenarios.
>
> We believe that both traditional algorithms and our method have their own merits and demerits. To illustrate this, we have included these discussions in the revised Appendix C.2.

---

> ### Author Response · Authors · 2025-11-27
>
> > **W3: RL Formulation.**
>
> We appreciate the reviewer’s insightful comments. Our work follows the standard formulation used in [1-2]. To improve clarity, we have added the full formulation in Section 2 of the revised manuscript.
>
> The multi-task VRP solver acts as a single agent, using the encoder-decoder architecture as its policy network. The policy generates a node sequence autoregressively, using a Markov Decision Process (MDP) environment
> \begin{equation}
>     \mathcal{M} = (\mathcal{S}, \mathcal{A}, \mathcal{P}, \mathcal{R}).
> \end{equation}
> **(1) State ($\mathcal{S}$)** consists of node embeddings and context embeddings. During decoding, following [1,2,6,7], the model explores from diverse starting points, forming multiple trajectories in parallel. Each trajectory maintains its own context (e.g., current time and used capacity), while all trajectories share the same set of node embeddings.
>
> **(2) Action ($\mathcal{A}$)** corresponds to selecting the next node to visit. The policy network takes the current state as input and generates a trajectory-specific probability distribution over feasible nodes, allowing each trajectory to independently select its next action based on the predicted probabilities.
>
> **(3) Transition ($\mathcal{P}$)** updates the environment after a node is selected. This modifies the environmental routing information, such as the vehicle’s current position and remaining capacity. The updated environment then defines the next context embedding and continues the decision process.
>
> **(4) Reward ($\mathcal{R}$)** is defined as the negative total route length. After all nodes are visited, each trajectory computes its own negative route length as the reward. These rewards, together with the action log-probabilities produced by the policy network, are aggregated to form a single training objective. The policy network parameters $\theta$ are then updated using the REINFORCE gradient [8]:
> \begin{equation}
> \nabla_\theta J(\theta) = \frac{1}{N} \sum_{i=1}^{N} (R_i - b) \nabla_\theta \log \pi_\theta(a_i \mid s_i),
> \end{equation}
> where $i$ is the index of the trajectory and $\pi_\theta(a_i \mid s_i)$ denotes the probability assigned to action $a_i$ conditioned on state $s_i$. $b$ is a shared baseline used to reduce gradient variance, computed as the average reward over all trajectories.
>
> > **W4: T-test Experiments.**
>
> Thank you very much for the constructive suggestions. While original Appendix C.1 has already provided an error-bar analysis, which evaluates the stability of CCL across different random seeds during inference, we continue to include t-tests to assess statistical significance. We first collected the gap values of 1,000 test instances from both CCL† and the strongest SOTA CaDA† from Table 1, then we report the improvement percentages along with the corresponding p-values as follows.
>
> |Task|CVRP|OVRP|OVRPB|OVRPBL|OVRPL|VRPB|VRPBL|VRPL|
> |----|----|----|----|----|----|----|----|----|
> |Improvement|8.16%|29.21%|29.86%|31.05%|28.69%|20.14%|20.03%|7.66%|
> |P_Value|2.7e-05|1.2e-76|1.4e-62|4.7e-68|3.5e-74|5.0e-42|2.0e-39|2.4e-04|
>
>
> |Task|OVRPBLTW|OVRPBTW|OVRPLTW|OVRPTW|VRPBLTW|VRPBTW|VRPLTW|VRPTW|
> |----|----|----|----|----|----|----|----|----|
> |Improvement|51.19%|50.63%|51.50%|50.85%|41.61%|46.84%|41.39%|45.72%|
> |P_Value|6.2e-68|2.8e-68|2.9e-79|3.2e-75|2.2e-68|4.0e-81|2.2e-97|2.9e-99|
>
> Here, the improvements are computed as the average gap reductions of CCL† over CaDA†, i.e., $-(Gap(CCL†)-Gap(CaDA†))/Gap(CaDA†)×100%$.
>
> Across all 16 tasks, CCL achieves 7–51% improvement. In particular, ovrpbltw, ovrpbtw, ovrpltw, and ovrptw exceed 50%. All p-values are below 0.001, indicating that these gains are statistically significant. In the revised manuscript, the original Appendix C.1 has been moved to Appendix C.1.1, and a new t-test analysis has been added in Appendix C.1.2.

---

> ### Author Response · Authors · 2025-11-27
>
> > **Q1: Feature Dimension.**
>
> **(1) Dimension of the Encoded Node Embeddings.**
>
> We refine the definition of the encoded node embeddings as
>
> \begin{equation}
>     \mathbf{H} = \mathcal{E}(\tilde{\mathbf{h}}, \mathbf{h}),
> \end{equation}
> where the inputs includes the contraint flag $\tilde{\mathbf{h}}$ and the node attributes $\mathbf{h}=( \mathbf{h}_0, \mathbf{h}_1,\dots, \mathbf{h}_N )$. The resulting feature dimension is $\mathbf{H} \in \mathbb{R}^{(N+1)\times D}$. Specifically, the constraint flag $\tilde{\mathbf{h}}$ is duplicated for each node, resulting in a corresponding embedding of dimension $(N+1)\times D$. Given $(N+1)\times D$ node embeddings, these two embeddings are concatenated along the feature dimension and projected back to dimension $D$, yielding the node embeddings of size $(N+1)\times D$.
>
> This procedure corresponds to processing the input attributes before the encoder, and it is described in the original Appendix B.1. For clarity, we have revised Section 3.2 and refer readers to Appendix B.1 for further details. The original Equation 1 has been renumbered as Equation 3 in the updated manuscript.
>
> **(2) Dimension of Concatenated Constraint Embeddings.**
>
> The concatenated embedding lies in $\mathbb{R}^{4D}$. Equation 5 in the original manuscript is given by (for clarity, the indices $i,j$ are omitted due to OpenReview limitations)
> \begin{equation}
>     \tilde{\mathbf{S}} = \mathcal{H}(\texttt{Concat}(\mathbf{C} ^{B}, \mathbf{C} ^{L}, \mathbf{C} ^{O}, \mathbf{C} ^{TW})),
> \end{equation}
>
>
>
> where $\mathbf{C}^{k} \in \mathbb{R}^{D}$ and $k \in ( B,L,O,TW)$.  $\texttt{Concat}(\cdot)$ denotes concatenation along the feature dimension, resulting in a concatenated embedding of size $4D$. $\mathcal{H}(\cdot)$ is a linear layer projecting the $4D$ input back to $D$, resulting in $\tilde{\mathbf{S}} \in \mathbb{R}^{D}$. For clarity, we have revised this explanation in Section 3.3, and the original Equation 5 is moved to Equation 7.
>
>
> > **Q2: Notation Explanation for N&L, and C&L**
>
> In Figure 1, N\&L and C\&L denote Normalization followed by a Linear layer, and Concatenation followed by a Linear layer, respectively. The original figure already included a legend. For clarity, we have revised the figure to use full names.
>
> > **Q3&Q4&Q6: Multiple Trajectories**
>
> **(1) Multiple Trajectories and Multi-Trajectory Context.**
>
> In Section 3.1, we state that the decoding stage explores from diverse starting points, thereby forming multiple trajectories in parallel. At a given step, each trajectory has its own actions, resulting in a distinct context (e.g., used capacity or current time). These contexts are concatenated to form the multi-trajectory context, which is an N×D embedding introduced at the beginning of Section 3.4.
>
> **(2) Definition of Trajectories and Demand Notation**
>
> During each decision step, the trajectory corresponds to a sub-route, i.e., the partial sequence of nodes visited so far. After all nodes are visited, the trajectory refers to the complete route, which is used to compute the final reward. $\delta_{\tau_{i,j}}^l, \delta_{\tau_{i,j}}^b$ refers to the linehaul and backhaul demands of the node selected for trajectory $i$ at decoding step $j$, where $\tau_{i,j}$ is the index of the selected node. Since each trajectory may select a different node at each step, the demands are indexed by $i$, $j$ to correspond to the selected nodes.
>
> ## References
>
> [1] Routefinder: Towards foundation models for vehicle routing problems. In TMLR, 2025.
>
> [2] Cada: Cross-problem routing solver with constraint-aware dual-attention. In ICML, 2025.
>
> [3] Pyvrp: A high-performance vrp solver package. In INFORMS Journal on Computing, 2024.
>
> [4] Gurobi Optimization. In https://www.gurobi.com, 2024.
>
> [5] An effective heuristic algorithm for the traveling-salesman problem. In Operations Research, 1973.
>
> [6] Multi-task learning for routing problem with cross-problem zero-shot generalization. In SIGKDD, 2024.
>
> [7] MVMoe: Multi-task vehicle routing solver with mixture-of-experts*. In ICML, 2024.
>
> [8] Simple statistical gradient-following algorithms for connectionist reinforcement learning. In Machine Learning, 1992.

---

### Author Response · Authors · 2025-12-01
**Final Remarks**

Dear Area Chair,

Thank you for the effort in reading our paper, our response, and the corresponding discussions.

This paper aims to develop **a unified neural network for solving 48 vehicle routing problem (VRP) variants (tasks)**. Existing methods overlook the dynamic nature of VRPs, where both **node urgencies and constraint priorities** change across decoding steps. Since these solvers explore solutions from multiple start points (trajectories), directly re-embedding nodes would cause computational overhead. To address these limitations, we propose CCL for **efficient dynamic decoding**. At each step, CCL incorporates RGCR to identify which constraints are more important, and applies TSNR to update shared nodes across multiple trajectories. This approach advances the SoTA across **16 in-distribution and 32 out-of-distribution VRP tasks**, particularly on complex tasks with time windows.

Reviewers broadly acknowledge this contribution, giving **scores of 6, 6, 6, 4, 4** and confidence ratings of 4, 4, 3, 3, 4. Our response was submitted at the time of the OpenReview bug. Only reviewer yziX has replied, confirming that we addressed most concerns while retaining a positive score. In the revised manuscript, we have addressed their concerns across various sections.

(1) In Section 1 (Introduction), we have explicitly illustrated our motivation from a Markov Decision Process (MDP) perspective, showing how CCL enables **a more accurate state construction** under the Markov property. (Reviewer jrJZ, Uhy6, and xK8d)

(2) In Section 2 (Preliminaries), we have provided an MDP formulation for enhanced readability. (Reviewer A3oC, jrJZ, yziX, and xK8d)

(3) In Section 4 (Experiments), we have conducted additional ablation studies, demonstrating that **CCL's improvements stem from its design rather than the network scale**. (Reviewer jrJZ and Uhy6)

(4) In Appendix C, we have added CCL's comparison with traditional solvers (Reviewer A3oC), significance analysis (Reviewer A3oC), ablations without ReLD (Reviewer Uhy6), and additional scenarios beyond coordinate-based VRPs (Reviewer yziX). Specifically, the comparison with traditional solvers shows that learning-based solvers perform better for **real-time applications**, while a unified model covering 48 tasks **broadens practical deployment scenarios**.

We hope that this summary will facilitate your final decision-making process. Thank you for your time and consideration.

Sincerely,

The Authors of Paper #1397

---

### Meta-Review · Area_Chair_e7ob · 2026-01-06

**Summary:**

This paper propose Chain-of-Context Learning (CCL), a framework that help neural solvers for Vehicle Routing Problems better handle dynamic constraints by updating node embeddings and prioritizing important constraints at each decoding step. By using the RGCR and TSNR modules, the model achieve state-of-the-art performance on 48 different VRP tasks, showing it can generalize to new, unseen problems.
Despite many nice contributions pointed out by reviewers, some of the stronger concerns that have been mentioned include:
1. some of the reviewers initially worried that the performance gains were "extremely incremental" and might only result from having more parameters in the network
2. There was concern about the "significant" increase in inference time compared to baselines, which might make the extra accuracy not worth the cost
3.  Multiple reviewers noted the paper did not have a clear Markov Decision Process (MDP) formulation, making it hard to understand how the reinforcement learning actually works
4. Some reviewers questioned the use of RL for these problems since traditional solvers like Gurobi or LKH can find optimal solutions for small scales
5. some reviewers argued that the authors' claim of capturing "historical preferences" might violate the Markov property, which is the basis for most RL

**Reviewer Concerns:**

# Addressed Concerns
*   The authors successfully demonstrated that the performance improvements are not just a product of scaling. They provided a "heavy-decoder" variant of the state-of-the-art (SOTA) baseline (CaDA-HD) which had more parameters than CCL, yet CCL still performed better. They also showed that the TSNR module adds zero additional parameters while significantly improving results.
*   In the revised Section 2 and their rebuttal, the authors provided a full, formal Markov Decision Process specification, including clear definitions for State, Action, Transition, and Reward (negative total route length). This clarified how the autoregressive decoding fits into a standard RL framework.
*  The authors conducted new experiments comparing CCL to Gurobi and LKH. While traditional solvers can find optimal solutions, the authors proved that CCL is 100x to 1200x faster, making it viable for real-time applications where a 15-minute Gurobi solve is not practical.
*   The authors clarified a misunderstanding regarding "historical preferences." They explained they are not using the full history to make decisions, but rather updating node embeddings at each step to ensure the State accurately reflects the current environment. This maintains the Markov property while improving the alignment between nodes and context.

# Outstanding Concerns
*  While the authors provided a "lightweight" version of CCL to show flexibility, the fact remains that CCL is slower than the current SOTA baselines (e.g., 6.5s vs 2.7s for 1,000 instances). Although they argue the accuracy gain justifies this, for users who prioritize raw speed above all else, the computational cost of the step-wise re-embedding is still a valid drawback.
* Some reviewers noted that the framework is very tailored to VRPs. While the authors added a small experiment on Flow Shop Problem (FFSP) to show generalizability, the method’s broader impact on non-routing combinatorial optimization remains less proven compared to its performance on VRP variants.

**Reviewer Scores:**

*  Reviewer yziX (initial score 6) - he seems to be happy with the rebuttal and most likely would keep or slightly increase.
*  Reviewer A3oC (initial score 4) - one of his worries was the lack of comparison with classical solvers like LKH or Gurobi. The new results are quite nice and he could be inclined to increase his score.
*  Reviewer Uhy6 (initial score 6) - was positive about the thorough evaluation. Because the authors directly answer their technical question with new data, this reviewer would likely keep their 6 or even move to a 8.
*  Reviewer xK8d (initial score 6) was very worry that the paper "violate the Markov property". The authors explain that they are not using the whole history, but just updating the state so it is more accurate. They even provide a new formal MDP formulation to prove it. I feel authors address the main weaknesses they pointed our and hence the reviewer would almost certainly stay at a 6 or slightly improve score.
*  Reviewer jrJZ (initial score 4) was the most skeptical about the "incremental" gains and the slow inference time. The authors show that the gains are not just from model size by comparing with other SOTA . I feel that the reviewer could increase the score after rebuttal.

---

### Decision · Program_Chairs · 2026-01-26

Accept (Poster)